# The Price of Implicit Bias in Adversarially Robust Generalization

**Nikolaos Tsilivis**[*]
New York University
nt2231@nyu.edu

**Natalie S. Frank**
New York University
nf1066@nyu.edu

**Nathan Srebro**
TTI-Chicago
nati@ttic.edu

**Julia Kempe**
New York University
Meta FAIR
kempe@nyu.edu

## Abstract

We study the implicit bias of optimization in robust empirical risk minimization (robust ERM) and its connection with robust generalization. In classification settings under adversarial perturbations with linear models, we study what type of regularization should ideally be applied for a given perturbation set to improve (robust) generalization. We then show that the implicit bias of optimization in robust ERM can significantly affect the robustness of the model and identify two ways this can happen; either through the optimization algorithm or the architecture. We verify our predictions in simulations with synthetic data and experimentally study the importance of implicit bias in robust ERM with deep neural networks.

## 1 Introduction

Robustness is a highly desired property of any machine learning system. Since the discovery of adversarial examples in deep neural networks [Szegedy et al., 2014, Biggio et al., 2013], adversarial robustness - the ability of a model to withstand small, adversarial, perturbations of the input at test time - has received significant attention. A canonical way to obtain a robust model $f$, parameterized by $\mathbf{w}$, is to optimize it for robustness during training, i.e. given a set of training examples $(\mathbf{x}_i, y_i)_{i=1}^m$, optimize the empirical, worst-case, loss $l$, where worst-case refers to a predefined threat model $\Delta(\cdot)$ which encodes our notion of proximity for the task:

$$\min_{\mathbf{w}} \frac{1}{m} \sum_{i=1}^{m} \max_{\mathbf{x}_i' \in \Delta(\mathbf{x}_i)} l\left(f(\mathbf{x}_i'; \mathbf{w}), y_i\right). \tag{1}$$

This method of *robust Empirical Risk Minimization* (robust ERM aka *adversarial training* [Madry et al., 2018]) has been the workhorse in deep learning for optimizing robust models in the past few years. However, despite the outstanding performance of deep networks in "standard" classification settings, the same networks under robust ERM lag behind; progress, in terms of absolute performance, has stagnated as measured on relevant benchmarks [Croce et al., 2021] and predicted by experimental scaling laws for robustness [Debenedetti et al., 2023], and any advances mainly rely on extreme amounts of synthetic data (see, e.g., [Wang et al., 2023]). Additionally, the (robust) generalization gap of neural networks obtained with robust ERM is large and, during training, networks typically exhibit overfitting [Rice et al., 2020]; (robust) test error goes up after initially going down, even though (robust) train error continues to decrease. How can we reconcile all this with the modern paradigm of deep learning, where overparameterized models interpolate their (even noisy) training data and seamlessly generalize to new inputs [Belkin, 2021]? What is different in robust ERM?

In "standard" classification, it is now understood that the optimization procedure is responsible for *capacity control* during ERM [Neyshabur et al., 2015] and this in turn permits generalization.

---

[*]Part of this work was done while author was with TTI-Chicago.

38th Conference on Neural Information Processing Systems (NeurIPS 2024).

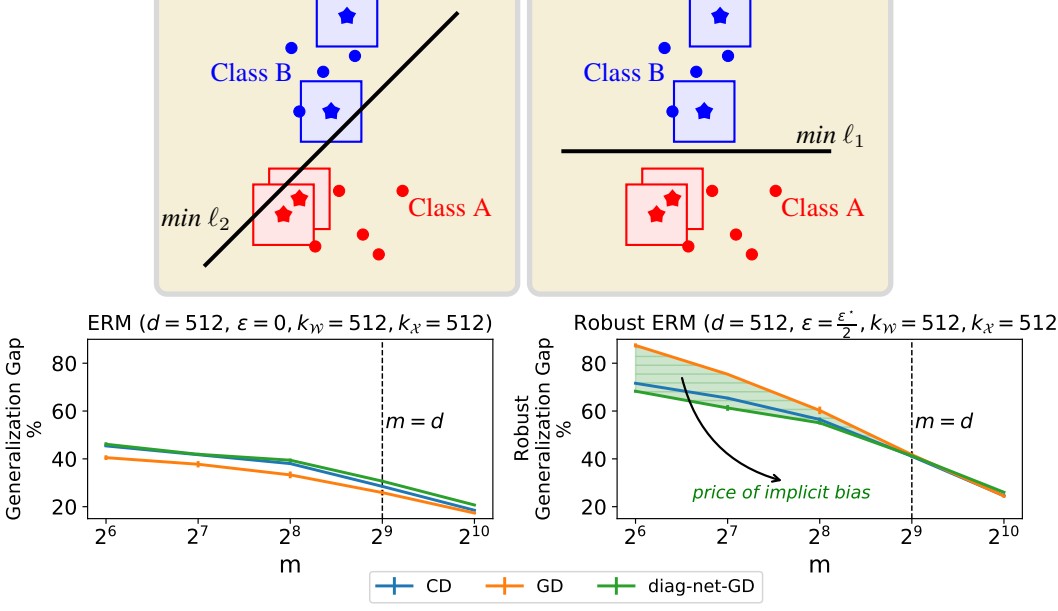

Figure 1: *The price of implicit bias* in adversarially robust generalization. **Top**: An illustration of the role of geometry in robust generalization: a separator that maximizes the $\ell_2$ distance between the training points (circles) might suffer a large error for test points (stars) perturbed within $\ell_\infty$ balls, while a separator that maximizes the $\ell_\infty$ distance might generalize better. **Bottom**: Binary classification of Gaussian data with (*right*) or without (*left*) $\ell_\infty$ perturbations of the input in $\mathbb{R}^d$ using linear models. We plot the (robust) generalization gap, i.e., (robust) train minus (robust) test accuracy, of different learning algorithms versus the training size $m$. In standard ERM ($\epsilon = 0$), the algorithms generalize similarly. In robust ERM, however, the implicit bias of gradient descent is hurting the robust generalization of the models, while the implicit bias of coordinate descent/gradient descent with diagonal linear networks aids it. See Section 4 for details.

We use the term capacity control to refer to the way that our algorithm imposes constraints on the hypotheses considered during learning; this can be achieved by means of either explicit (e.g. weight decay [Krogh and Hertz, 1991]) or implicit regularization [Neyshabur et al., 2015] induced by the optimization algorithm [Soudry et al., 2018, Gunasekar et al., 2018a], the loss function [Gunasekar et al., 2018a], the architecture [Gunasekar et al., 2018b] and more. This implicit bias of optimization towards empirical risk minimizers with small capacity (some kind of "norm") is what allows them to generalize, even in the absence of explicit regularization [Zhang et al., 2017], and can, at least partially, explain why gradient descent returns well-generalizing solutions [Soudry et al., 2018].

**Our contributions** In this work, we explore the implicit bias of optimization in **robust** ERM and study carefully how it affects the **robust generalization** of a model. In order to overcome the hurdles of the bilevel optimization in the definition of robust ERM (eq. (1)), we seek to first understand the situation in linear models, where the inner minimization problem admits a closed form solution. Prior work [Yin et al., 2019, Awasthi et al., 2020] that studied generalization bounds for this class of models for $\ell_p$ norm-constrained perturbations observed that the hypothesis class (class of linear predictors) should better be constrained in its $\ell_r$ norm with $r$ smaller than or equal to $p^\star$, where $p^\star$ is the dual of the perturbation norm $p$, i.e $\frac{1}{p} + \frac{1}{p^\star} = 1$. For instance, in the case of $\ell_\infty$ perturbations, these works postulated that searching for robust empirical risk minimizers with small $\ell_1$ norm is beneficial for robust generalization. In Section 2, we further refine these arguments and demonstrate that there are also other factors, namely the sparsity of the data and the magnitude of the perturbation, which can influence the choice of the regularizer norm $r$. Nevertheless, in accordance with [Yin et al., 2019, Awasthi et al., 2020], we do identify cases where insisting on a suboptimal type of regularization makes generalization more difficult - much more difficult than in "standard" classification.

This observation has significant implications for training robust models. The hidden gift of optimization that allowed generalization in the context of ERM can now become a punishment in robust ERM, if implicit bias and threat model happen to be "misaligned" with each other. We call this the *price of implicit bias* in adversarially robust generalization and demonstrate two ways this price can appear; either by varying the optimization algorithm or the architecture. In particular, we first focus on robust ERM over the class of linear functions $f_{\text{lin}}(\mathbf{x}; \mathbf{w}) = \langle \mathbf{w}, \mathbf{x} \rangle$ with *steepest descent* with respect to an $\ell_r$ norm, a class of algorithms which generalizes gradient descent to other geometries besides the Euclidean (Section 3.1). In the case of separable data, we prove that robust ERM with infinitesimal step size with the exponential loss asymptotically reaches a solution with minimum $\ell_r$ norm that classifies the training points robustly (Theorem 3.3). Although this result is to be expected, given that standard ERM with steepest descent also converges to a minimum norm solution (without the robustness constraint, however) [Gunasekar et al., 2018a], it lets us argue that, in certain cases, gradient descent-based robust ERM will generalize poorly *despite* the existence of better alternatives - see Figure 1 (bottom). We then turn our attention to study the role of architecture in robust ERM. In Section 3.2, we study the implicit bias of gradient descent-based robust ERM in models of the form $f_{\text{diag}}(\mathbf{x}; \mathbf{u}_+, \mathbf{u}_-) = \langle \mathbf{u}_+^2 - \mathbf{u}_-^2, \mathbf{x} \rangle$, commonly referred to as *diagonal neural networks* [Woodworth et al., 2020]. These are just reparameterized linear models $f_{\text{lin}}$, and thus their expressive power does not change. Yet, as we show, robust ERM drives them to solutions with very different properties (Proposition 3.8) than those of $f_{\text{lin}}$, which can generalize robustly much better - see Figure 1 (bottom).

Finally, in Section 4, we perform extensive simulations with linear models over synthetic data which illustrate the theoretical predictions and, then, investigate the importance of implicit bias in robust ERM with deep neural networks over image classification problems. In analogy to situations we encountered in linear models, we find evidence that the choice of the algorithm and the induced implicit bias affect the final robustness of the model more and more as the magnitude of the perturbation increases.

**Notation**    Let $[m] = \{1, \ldots, m\}$. The dual norm of a vector $\mathbf{z}$ is defined as $\|\mathbf{z}\|_\star = \sup_{\|\mathbf{x}\| \leq 1} \langle \mathbf{z}, \mathbf{x} \rangle$. The dual of an $\ell_p$ norm is the $\ell_{p^\star}$ with $\frac{1}{p} + \frac{1}{p^\star} = 1$. For a function $f : \mathbb{R}^d \to \mathbb{R}$, we use $\partial f(\mathbf{x})$ to denote the set of subgradients of $f$ at $\mathbf{x}$: $\partial f(\mathbf{x}) = \{ \mathbf{g} \in \mathbb{R}^d : f(\mathbf{z}) \geq f(\mathbf{x}) + \langle \mathbf{g}, \mathbf{z} - \mathbf{x} \rangle \}$. We denote by $\mathbf{x}^2 \in \mathbb{R}^d$ the element-wise square of $\mathbf{x}$. We defer a full discussion of prior work to App. A.

## 2   Capacity Control in Adversarially Robust Classification

We begin by studying the connection between *explicit* regularization and *robust generalization* error in linear models. In particular, we set out to understand how constraining the $\ell_r$ norm of a model affects its robustness with respect to $\ell_p$ norm perturbations, i.e., how $r$ interacts with $p$.

### 2.1   Generalization Bounds for Adversarially Robust Classification

We focus on binary classification with linear models over examples $\mathbf{x} \in \mathcal{X} \subseteq \mathbb{R}^d$ and labels $y \in \{\pm 1\}$. We denote by $\mathcal{D}$ an unknown distribution over $\mathcal{X} \times \{\pm 1\}$. We assume access to $m$ pairs from $\mathcal{D}$, $S = \{(\mathbf{x}_1, y_1), \ldots, (\mathbf{x}_m, y_m)\}$. Let $\mathcal{H}_r$ be the class of linear hypotheses with a restricted $\ell_r$ norm:

$$\mathcal{H}_r = \{\mathbf{x} \mapsto \langle \mathbf{w}, \mathbf{x} \rangle : \|\mathbf{w}\|_r \leq \mathcal{W}_r\}, \tag{2}$$

where $\mathcal{W}_r > 0$ is an arbitrary upper bound. We consider loss functions of the form $l(h(\mathbf{x}), y) = l(yh(\mathbf{x}))$, by explicitly overloading the notation with $l : \mathbb{R} \to [0, 1]$. The quantity $yh(\mathbf{x})$ is sometimes referred to as the *confidence margin* of $h$ on $(\mathbf{x}, y)$. We assume a threat model of $\ell_p$ balls of radius $\epsilon$ centered around the original samples and we define $\mathcal{G}_r$ to be the class of functions that map samples to their worst-case loss value, i.e. $\mathcal{G}_r = \{(\mathbf{x}, y) \mapsto \max_{\|\mathbf{x}' - \mathbf{x}\|_p \leq \epsilon} l(yh(\mathbf{x}')) : h \in \mathcal{H}_r\}$. We define the (expected) risk and empirical risk of a hypothesis with respect to the worst-case loss as:

$$\widetilde{L}_\mathcal{D}(h) = \mathbb{E}_{(\mathbf{x}, y) \sim \mathcal{D}} \left[ \max_{\|\mathbf{x}' - \mathbf{x}\|_p \leq \epsilon} l(yh(\mathbf{x}')) \right] \text{ and } \widetilde{L}_S(h) = \frac{1}{m} \sum_{i=1}^m \max_{\|\mathbf{x}'_i - \mathbf{x}_i\|_p \leq \epsilon} l\left(y_i h(\mathbf{x}'_i)\right), \tag{3}$$

respectively.      Let us also define the robust 0-1 risk as: $\widetilde{L}_{\mathcal{D},01}(h) = \mathbb{E}_{(\mathbf{x}, y) \sim \mathcal{D}} \left[ \max_{\|\mathbf{x}' - \mathbf{x}\|_p \leq \epsilon} \mathbb{1}\{yh(\mathbf{x}') \leq 0\} \right]$.    Central to the analysis of the robust generaliza-

tion error is the notion of the (empirical) Rademacher Complexity of the function class $\mathcal{G}_r$:

$$\hat{\mathfrak{R}}_S(\mathcal{G}_r) = \mathbb{E}_\sigma \left[ \frac{1}{m} \sup_{g \in \mathcal{G}_r} \sum_{i=1}^m \sigma_i g((x_i, y_i)) \right] = \mathbb{E}_\sigma \left[ \frac{1}{m} \sup_{h \in \mathcal{H}_r} \sum_{i=1}^m \sigma_i \max_{\|\mathbf{x}_i' - \mathbf{x}_i\|_p \le \epsilon} l(y_i h(\mathbf{x}_i')) \right], \quad (4)$$

where the $\sigma_i$'s are Rademacher random variables. If, additionally, we consider decreasing, Lipschitz, losses $l(\cdot)$, then, as observed by Yin et al. [2019], Awasthi et al. [2020], we can equivalently analyse the following Rademacher Complexity $\hat{\mathfrak{R}}_S(\widetilde{\mathcal{H}}_r) = \mathbb{E}_\sigma \left[ \frac{1}{m} \sup_{h \in \mathcal{H}_r} \sum_{i=1}^m \sigma_i \min_{\|\mathbf{x}_i' - \mathbf{x}_i\|_p \le \epsilon} y_i h(\mathbf{x}_i') \right]$, and by taking the loss in $\widetilde{L}_\mathcal{D}(\cdot), \widetilde{L}_S(\cdot)$ in eq. (3) to be the *ramp loss*: $l(u) = \min \left( 1, \max \left( 0, 1 - \frac{u}{\rho} \right) \right)$, $\rho > 0$, we arrive at the following margin-based generalization bound.

**Theorem 2.1.** *[Mohri et al., 2012, Awasthi et al., 2020] Fix $\rho > 0$. For any $\delta > 0$, with probability at least $1 - \delta$ over the draw of the dataset $S$, for all $h \in \mathcal{H}_r$ with $\mathcal{H}_r$ defined as in eq. (2), it holds:*

$$\widetilde{L}_\mathcal{D}(h) \le \widetilde{L}_S(h) + \frac{2}{\rho} \hat{\mathfrak{R}}_S(\widetilde{\mathcal{H}}_r) + 3\sqrt{\frac{\log 2/\delta}{2m}}. \quad (5)$$

Margin bounds of this kind are attractive, since they promise that, if the empirical margin risk $\widetilde{L}_S$ is small for a large $\rho$ then the second term in the RHS will shrink, and expected and empirical risk will be close. As shown in [Awasthi et al., 2020], the above Rademacher complexity admits an upper bound (and a matching lower bound) of the form:

$$\hat{\mathfrak{R}}_S(\widetilde{\mathcal{H}}_r) \le \hat{\mathfrak{R}}_S(\mathcal{H}_r) + \epsilon \frac{\mathcal{W}_r}{2\sqrt{m}} \max \left( d^{\frac{1}{p^\star} - \frac{1}{r}}, 1 \right), \quad (6)$$

where $\hat{\mathfrak{R}}_S(\mathcal{H}_r)$ is the "standard" Rademacher complexity. As pointed out in [Awasthi et al., 2020], there is a dimension dependence appearing in this bound, that is not present in the "standard" case of $\epsilon = 0$, and, thus, it makes sense to choose $r$ so that we eliminate that term. One such choice is of course $r = p^\star$, the dual of $p$. This made the works of Yin et al. [2019], Awasthi et al. [2020] to advocate for an $\ell_{p^\star}$ regularization during training, in order to minimize the complexity term and, hence, the robust generalization error. However, the factor $\mathcal{W}_r$ that appears in the RHS of eq. (6) might also depend on $r$ (and potentially $d$) so it is not entirely clear what the optimal choice of $r$ is for a problem at hand.

## 2.2 Optimal Regularization Depends on Sparsity of Data

To illustrate the previous point, we place ourselves in the realizable setting, where there exists a linear "teacher" which labels the samples *robustly*. That is, there is a vector $\mathbf{w}^\star \in \mathbb{R}^d$ which labels points and their neighbors with the same label: $y = \text{sgn}(\langle \mathbf{w}^\star, \mathbf{x}' \rangle)$ for all $\mathbf{x}' \in \{\mathbf{z} \in \mathbb{R}^d : \|\mathbf{z} - \mathbf{x}\|_p \le \epsilon\}$. Let us specialize to hypothesis classes with bounded $\ell_1$ or $\ell_2$ norm, i.e. $\mathcal{H}_1, \mathcal{H}_2$. Since the data are assumed to be labeled by a robust "teacher", the robust empirical risk that corresponds to the ramp loss can be driven to zero with a sufficiently large hypothesis class. The next Proposition provides a bound on the robust generalization of predictors who belong to such a class.

**Proposition 2.2.** *(Generalization bound for robust interpolators) Consider a distribution $\mathcal{D}$ over $\mathbb{R}^d \times \{\pm 1\}$ with $\mathbb{P}_{(\mathbf{x},y) \sim \mathcal{D}} [y = \text{sgn}(\langle \mathbf{w}^\star, \mathbf{x} \rangle), \forall \mathbf{x}' : \|\mathbf{x}' - \mathbf{x}\|_p \le \epsilon] = 1$ for some $\mathbf{w}^\star \in \mathbb{R}^d$. Let $S \sim \mathcal{D}^m$ be a draw of a random dataset $S = \{(\mathbf{x}_1, y_1), \ldots, (\mathbf{x}_m, y_m)\}$ and let $\mathcal{H}_r' = \left\{ \mathbf{x} \mapsto \langle \mathbf{w}, \mathbf{x} \rangle : \|\mathbf{w}\|_r \le \|\mathbf{w}^\star\|_r \wedge \mathbf{w} \in \text{argmax}_{\|\mathbf{u}\|_r \le 1} \min_{i \in [m]} \min_{\|\mathbf{x}_i' - \mathbf{x}\|_p \le \epsilon} y_i \langle \mathbf{u}, \mathbf{x}_i' \rangle \right\}$ be a hypothesis class of maximizers of the robust margin. Then, for any $\delta > 0$, with probability at least $1 - \delta$ over the draw of the random dataset $S$, for all $h \in \mathcal{H}_r'$, it holds:*

$$\widetilde{L}_{\mathcal{D},01}(h) \le \begin{cases} \frac{1}{2\sqrt{m}} \left( \max_i \|\mathbf{x}_i\|_\infty \|\mathbf{w}^\star\|_1 \sqrt{2 \log(2d)} + \epsilon \|\mathbf{w}^\star\|_1 \right) + 3\sqrt{\frac{\log 2/\delta}{2m}}, & r = 1 \\ \frac{1}{2\sqrt{m}} \left( \max_i \|\mathbf{x}_i\|_2 \|\mathbf{w}^\star\|_2 + \epsilon \|\mathbf{w}^\star\|_2 d^{\max\left(\frac{1}{p^\star} - \frac{1}{2}, 0\right)} \right) + 3\sqrt{\frac{\log 2/\delta}{2m}}, & r = 2. \end{cases} \quad (7)$$

The proof appears in Appendix B and follows from standard arguments based on the properties of the ramp loss and standard Rademacher complexity bounds. Notice that eq. (7) depends on the various norms of $\mathbf{w}^\star$ and $\mathbf{x}$, so we can consider specific cases in order to probe its behaviour in different

regimes. In particular, we assume that all the entries of the vectors are normalized to be $\mathcal{O}(1)$. We call a vector $\mathbf{z} \in \mathbb{R}^d$ "dense" when it satisfies $\|\mathbf{z}\|_1 = \Theta(d)$ and $\|\mathbf{z}\|_2 = \Theta(\sqrt{d})$, while we call it "$k$-sparse" if $\|\mathbf{z}\|_1 = \Theta(k)$ and $\|\mathbf{z}\|_2 = \Theta(\sqrt{k})$ for $k < d$. Let us also specialize to $p = \infty$. We enumerate the cases:

1. *Dense, Dense*: If both the ground truth vector $\mathbf{w}^\star$ and the samples $\mathbf{x}$ (with probability 1) are dense, then the bounds evaluate to $\Theta\left(\frac{1}{\sqrt{m}}(d\sqrt{\log d} + \epsilon d)\right)$ and $\Theta\left(\frac{1}{\sqrt{m}}(d + \epsilon d)\right)$ for $r = 1$ and $r = 2$, respectively. In particular, for $\epsilon = 0$, the $r = 2$ bound is smaller only by a logarithmic factor, and as $\epsilon$ increases the bounds should behave the same. So, we expect an $\ell_2$ regularization to yield smaller generalization error for $\epsilon = 0$, while for larger $\epsilon$, $\ell_2$ and $\ell_1$ regularization should perform roughly similarly.

2. *$k$-Sparse, Dense*: If the ground truth vector is $k$-sparse and the samples are dense, then the bounds yield $\Theta\left(\frac{1}{\sqrt{m}}(k\sqrt{\log d} + \epsilon k)\right)$ and $\Theta\left(\frac{1}{\sqrt{m}}(\sqrt{d}\sqrt{k} + \epsilon\sqrt{k}\sqrt{d})\right)$ for $r = 1$ and $r = 2$, respectively. For $k = \mathcal{O}(1)$, $\ell_1$ regularization is expected to generalize better than $\ell_2$ already for $\epsilon = 0$. As $\epsilon$ increases, $\ell_2$ regularized solutions should continue generalizing worse, as the "worst-case" dimension-dependent term makes its appearance.

3. *Dense, $k$-Sparse*: If $\mathbf{w}^\star$ is dense and the samples $\mathbf{x}$ are $k$-sparse, then we get $\Theta\left(\frac{1}{\sqrt{m}}(d\sqrt{\log d} + \epsilon d)\right)$ and $\Theta\left(\frac{1}{\sqrt{m}}(\sqrt{k}\sqrt{d} + \epsilon d)\right)$ for $r = 1$ and $r = 2$, respectively. The $r = 2$ bounds provides more favorable guarantees in this case, even for $\epsilon > 0$.

4. *$k$-Sparse, $k$-Sparse*: If both $\mathbf{w}^\star$ and $\mathbf{x}$ are $k$-sparse, then we have $\Theta\left(\frac{1}{\sqrt{m}}(k\sqrt{\log d} + \epsilon k)\right)$ and $\Theta\left(\frac{1}{\sqrt{m}}(k + \epsilon\sqrt{k}\sqrt{d})\right)$ for $r = 1$ and $r = 2$, respectively. For $\epsilon = 0$, $\ell_1$ and $\ell_2$ regularization should behave similarly, but, as $\epsilon$ increases, $\ell_2$ regularization starts "paying" the "worst-case" dimension-dependent term, making the $\ell_1$ solution more appealing.

Notice how the "price" of robustness especially manifests itself in Case 4, where our input is "embedded" in a $k$-dimensional space: the bounds are very similar for $\epsilon = 0$, but as soon as $\epsilon$ becomes positive, the extra penalty of $\ell_2$ solutions over $\ell_1$ grows with dimension. Moreover, Case 3 highlights that an $\ell_1$ regularization is not always optimal for $\ell_\infty$ perturbations. To summarize, we see that the optimal choice of regularization depends not only on the choice of norm $p$ and the value of $\epsilon$, but also on the sparsity of the data-generating process (see also Table 1 for a summary). In particular, in order for the dimension-dependent term to appear in the $r = 2$ bound, the model $\mathbf{w}^\star$ itself needs to be sparse.

## 3 Implicit Biases in Robust ERM

In the previous section, we saw that the way we choose to constrain our hypothesis class can significantly affect the robust generalization error. In this section, we connect this with the implicit bias of optimization during robust ERM and demonstrate cases where the implicit regularization is either working in favor of robust generalization or against it. The term implicit bias refers to the tendency of optimization methods to infuse their solutions with properties that were not explicitly "encoded" in the loss function. It usually describes the asymptotic behavior of the algorithm. We study two ways that an implicit bias can affect robustness in robust ERM: through the optimization algorithm and through the parameterization of the model.

### 3.1 Price of Implicit Bias from the Optimization Algorithm

In this section, we study the implicit bias of robust ERM in linear models with *steepest descent*, a family of algorithms which generalizes gradient descent to other than the Euclidean geometries. We focus on minimizing the worst-case exponential loss, which has the same asymptotic properties as the logistic or cross-entropy loss (see e.g. [Telgarsky, 2013, Soudry et al., 2018, Lyu and Li, 2020]):

$$\widetilde{L}_S(h) := \widetilde{L}_S(\mathbf{w}) = \sum_{i=1}^{m} \max_{\|\mathbf{x}'_i - \mathbf{x}_i\|_p \leq \epsilon} \exp(-y_i \langle \mathbf{w}, \mathbf{x}'_i \rangle) = \sum_{i=1}^{m} \exp(-y_i \langle \mathbf{w}, \mathbf{x}_i \rangle + \epsilon \|\mathbf{w}\|_{p^\star}). \quad (8)$$

The above corresponds to choosing $l(u) = \exp(-u)$ in the definition of eq. (3). We first proceed with some definitions about the margin and the separability of a dataset.

**Definition 3.1.** We call $\ell_p$-*margin* of a dataset $\{\mathbf{x}_i, y_i\}_{i=1}^m$ the quantity $\max_{\mathbf{w} \neq 0} \min_{i \in [m]} \frac{y_i \langle \mathbf{w}, \mathbf{x}_i \rangle}{\|\mathbf{w}\|_{p^\star}}$.

**Definition 3.2.** A dataset $\{\mathbf{x}_i, y_i\}_{i=1}^m$ is $(\epsilon, p)$-*linearly separable* if $\max_{\mathbf{w} \neq 0} \min_{i \in [m]} \frac{y_i \langle \mathbf{w}, \mathbf{x}_i \rangle}{\|\mathbf{w}\|_{p^\star}} \geq \epsilon$.

Geometrically, the $\ell_p$-margin of a dataset captures the largest possible $\ell_p$-distance of a decision boundary to their closest data point $\mathbf{x}_i$ (see Lemma C.8 for completeness). Requiring separability is a natural starting point for understanding training methods that succeed in fitting their training data and has been widely adopted in prior work [Soudry et al., 2018, Li et al., 2020, Lyu and Li, 2020]).

**Steepest Descent** *(Normalized)* steepest descent is an optimization method which updates the variables with a vector which has unit norm, for some choice of norm, and aligns maximally with minus the gradient of the objective function [Boyd and Vandenberghe, 2014]. Formally, the update for normalized steepest descent with respect to a norm $\|\cdot\|$ for a loss $L_S(\mathbf{w})$ is given by:

$$\mathbf{w}_{t+1} = \mathbf{w}_t + \eta_t \Delta \mathbf{w}_t, \text{ where } \Delta \mathbf{w}_t \text{ satisfies}$$
$$\Delta \mathbf{w}_t \in \underset{\|\mathbf{u}\| \leq 1}{\operatorname{argmin}} \langle \mathbf{u}, \nabla L_S(\mathbf{w}_t) \rangle. \tag{9}$$

Unnormalized steepest descent, or simply steepest descent, further scales the magnitude of the update by $\|\nabla L_S(\mathbf{w}_t)\|_\star$, where $\|\cdot\|_\star$ denotes the dual norm of $\|\cdot\|$. The case $\|\cdot\| = \|\cdot\|_2$ corresponds to familiar gradient descent. We will be interested in understanding the steepest descent trajectory, when minimizing $\widetilde{L}_S$ from eq. (8), in the limit of infinitesimal stepsize, i.e. steepest flow dynamics:

$$\frac{d\mathbf{w}}{dt} \in \left\{ \mathbf{v} \in \mathbb{R}^d : \mathbf{v} \in \underset{\mathbf{u} \in \mathbb{R}^d : \|\mathbf{u}\| \leq \|\mathbf{g}\|_\star}{\operatorname{argmin}} \langle \mathbf{u}, \mathbf{g} \rangle, \mathbf{g} \in \partial \widetilde{L}_S \right\}. \tag{10}$$

Notice that loss $\widetilde{L}_S$ is not differentiable everywhere (due to the norm in the exponent), but we can consider subgradients $\partial \widetilde{L}_S$ in our analysis. We are ready to state our result for the asymptotic behavior of steepest flow in minimizing the worst-case exponential loss.

**Theorem 3.3.** *For any $(\epsilon, p)$-linearly separable dataset and any initialization $\mathbf{w}_0$, consider steepest flow with respect to the $\ell_r$ norm, $r \geq 1$, on the worst-case exponential loss $\widetilde{L}_S(\mathbf{w}) = \sum_{i=1}^m \max_{\|\mathbf{x}_i' - \mathbf{x}_i\|_p \leq \epsilon} \exp(-y_i \langle \mathbf{w}, \mathbf{x}_i' \rangle)$. Then, the iterates $\mathbf{w}_t$ satisfy:*

$$\lim_{t \to \infty} \min_i \min_{\|\mathbf{x}_i' - \mathbf{x}_i\|_p \leq \epsilon} \frac{y_i \langle \mathbf{w}_t, \mathbf{x}_i' \rangle}{\|\mathbf{w}_t\|_r} = \max_{\mathbf{w} \neq 0} \min_i \min_{\|\mathbf{x}_i' - \mathbf{x}_i\|_p \leq \epsilon} \frac{y_i \langle \mathbf{w}, \mathbf{x}_i' \rangle}{\|\mathbf{w}\|_r}. \tag{11}$$

Theorem 3.3 can be seen as a generalization of the results of Gunasekar et al. [2018a] to robust ERM (for any $\ell_p$ perturbation norm), modulo our continuous time analysis. The choice of analyzing continuous-time dynamics was made to avoid many technical issues related to the non-differentiability of the norm, which do not affect the asymptotic behavior of the algorithm. Li et al. [2020] studied the implicit bias of gradient descent in robust ERM, and showed that it converges to the minimum $\ell_2$ solution that classifies the training points robustly, which agrees with the special case of $r = 2$ in Theorem 3.3. For the proof, we need to lower bound the margin at all times $t$ with a quantity that asymptotically goes to the maximum margin. This requires a *duality* lemma that relates the (sub) gradient of the loss with the maximum margin, and generalizes previous results that only apply to either gradient descent, or the unperturbed loss, but not to both. The proof appears in Appendix C.

*Remark* 3.4. The right hand side of eq. (11) is equivalent to:

$$\min_{\mathbf{w}} \|\mathbf{w}\|_r \quad \text{s.t.} \quad \min_{\|\mathbf{x}_i' - \mathbf{x}_i\|_p \leq \epsilon} y_i \langle \mathbf{w}, \mathbf{x}_i' \rangle \geq 1, \ \forall i \in [m]. \tag{12}$$

Thus, the solution converges, in direction, to the hyperplane with the smallest $\ell_r$ norm which classifies the training points correctly (and robustly). As a result, we can leverage Proposition 2.2 to reason about the robust generalization of the solution returned by steepest descent. An equivalent viewpoint of (12), first observed by Li et al. [2020] about a version of this result for gradient descent ($r = 2$), is the following:

$$\min_{\mathbf{w}} \|\mathbf{w}\|_r + \lambda(\epsilon, m)\|\mathbf{w}\|_{p^\star} \quad \text{s.t. } y_i \langle \mathbf{w}, \mathbf{x}_i \rangle \geq 1, \ \forall i \in [m], \tag{13}$$

for some $\lambda(\epsilon, m) > 0$. Thus, the problem can be understood as performing norm minimization for a norm which is a linear combination of the algorithm norm $r$ and the dual of the perturbation norm $p^\star$. The coefficient of the latter increases with $\epsilon$, which, hereby, means that the bias induced from the perturbation starts to dominate over the bias of the algorithm with increasing $\epsilon$.

In light of Section 2 and Proposition 2.2, we see that the implications of this result are twofold. First, on the negative side, Theorem 3.3 implies that robust ERM with gradient descent ($\ell_2$) can harm the robust generalization error if $p = \infty$. For instance, as we saw in Cases 2 and 4 in Section 2.2, gradient descent will suffer dimension dependent statistical overheads. On the positive side, Theorem 3.3 supplies us with an algorithm that can achieve the desired regularization. In Cases 2 and 4 this would correspond to steepest descent with respect to $r = 1$. In general, we have the following corollary:

**Corollary 3.5.** *Minimizing the loss of eq. (8) with steepest flow with respect to the $\ell_{p^\star}$ norm (on $(\epsilon, p)$ separable data) convergences to a minimum $\ell_{p^\star}$ norm solution that classifies all the points correctly.*

The notable case of steepest descent w.r.t. the $\ell_1$ norm is called *coordinate descent*. It amounts to updating at each step only the coordinate that corresponds to the largest absolute value of the gradient (Appendix D). In Section 4.1, we demonstrate how robust ERM w.r.t. $\ell_\infty$ perturbations with coordinate descent, can enjoy much smaller robust generalization error than gradient descent.

Finally, although the perturbation magnitude $\epsilon$ did not influence the conversation so far in terms of the choice of the algorithm, it is important to note that, as $\epsilon$ increases, the max-margin solution will look similar for any choice of norm. In fact, in the limiting case of the largest possible $\epsilon$ that does not violate the separability assumption, all max-margin separators are the same - see Lemma C.7 - so the type of implicit bias will cease to be important for generalization.

## 3.2 Price of Implicit Bias from Parameterization

We reasoned in the previous section that robust ERM with gradient descent over the class of linear functions of the form $f_{\text{lin}}(\mathbf{x}; \mathbf{w}) = \langle \mathbf{w}, \mathbf{x} \rangle$ can result in excessive (robust) test error for $\ell_\infty$ perturbations. We now demonstrate how the same algorithm, but applied to a *different architecture*, can induce much more robust models. In particular, consider the following architecture:

$$f_{\text{diag}}(\mathbf{x}; \mathbf{u}) = \left\langle \mathbf{u}_+^2 - \mathbf{u}_-^2, \mathbf{x} \right\rangle, \mathbf{u} = [\mathbf{u}_+, \mathbf{u}_-] \in \mathbb{R}^{2d}, \mathbf{x} \in \mathbb{R}^d, \tag{14}$$

which consists of a reparameterization of $f_{\text{lin}}$. In terms of expressive power, the two architectures are the same. However, optimizing them can result in very different predictors. In fact, this class of homogeneous models, known as *diagonal linear networks*, have been the subject of case studies before for understanding feature learning in deep networks, because, whilst linear in the input, they can exhibit non-trivial behaviors of feature learning [Woodworth et al., 2020]. In order to study the implicit bias of robust ERM with gradient descent on $f_{\text{diag}}$, we leverage a result by [Lyu and Zhu, 2022] which shows that, under certain conditions, the implicit bias of gradient flow based robust ERM for homogeneous networks, is towards solutions with small $\ell_2$ norm.

**Theorem 3.6** (Paraphrased Theorem 5 in [Lyu and Zhu, 2022]). *Consider gradient flow minimizing a worst-case exponential loss $\widetilde{L}_S(\mathbf{u}) = \frac{1}{m} \sum_{i=1}^m \max_{\|\mathbf{x}_i - \mathbf{x}_i'\|_p \leq \epsilon} e^{-y_i f(\mathbf{x}_i'; \mathbf{u})}$, for a homogeneous, locally Lipschitz, network $f(\mathbf{x}; \cdot) : \mathbb{R}^p \to \mathbb{R}$, and assume that for all times $t > 0$ and for each point $\mathbf{x}_i$ the perturbation $\arg\max_{\|\mathbf{x}_i - \mathbf{x}_i'\|_p \leq \epsilon} e^{-y_i f(\mathbf{x}_i'; \mathbf{u})}$ is scale invariant and that the loss gets minimized, i.e. $\widetilde{L}_S(\mathbf{u}) \overset{t \to \infty}{\to} 0$. Then, $\mathbf{u}$ converges in direction to a KKT point of the following optimization problem:*

$$\min_{\mathbf{u}} \frac{1}{2} \|\mathbf{u}\|_2^2 \quad s.t. \quad \min_{\|\mathbf{x}_i' - \mathbf{x}_i\|_p \leq \epsilon} y_i f(\mathbf{x}_i'; \mathbf{u}) \geq 1, \ \forall i \in [m]. \tag{15}$$

As we show, $f_{\text{diag}}$ satisfies the conditions of Theorem 3.6, and, thus, we get the following description of its asymptotic behavior.

**Corollary 3.7.** *Consider gradient flow on the worst-case exponential loss $\widetilde{L}_S(\mathbf{u}) = \frac{1}{m} \sum_{i=1}^m \max_{\|\mathbf{x}_i - \mathbf{x}_i'\|_p \leq \epsilon} e^{-y_i f_{\text{diag}}(\mathbf{x}_i'; \mathbf{u})}$ and assume that $\widetilde{L}_S(\mathbf{u}) \to 0$. Then, $\mathbf{u}$ converges in direction to a KKT point of the following optimization problem:*

$$\min_{\mathbf{u}_+ \in \mathbb{R}^d, \mathbf{u}_- \in \mathbb{R}^d} \frac{1}{2} \left( \|\mathbf{u}_+\|_2^2 + \|\mathbf{u}_-\|_2^2 \right) \quad s.t. \quad \min_{\|\mathbf{x}_i' - \mathbf{x}_i\|_p \leq \epsilon} y_i \left\langle \mathbf{u}_+^2 - \mathbf{u}_-^2, \mathbf{x}_i' \right\rangle \geq 1, \ \forall i \in [m]. \tag{16}$$

However, the optimization problem of eq. (16), which is over $\mathbb{R}^{2d}$ is nothing but a disguised $\ell_1$ minimization problem, when viewed in the *prediction* ($\mathbb{R}^d$) space.

**Proposition 3.8.** *Problem* (16) *has the same optimal value as the following constrained opt. problem:*

$$\min_{\mathbf{w}\in\mathbb{R}^d} \|\mathbf{w}\|_1 \quad s.t. \quad \min_{\|\mathbf{x}_i'-\mathbf{x}_i\|_p \leq \epsilon} y_i \langle \mathbf{w}, \mathbf{x}_i' \rangle \geq 1, \ \forall i \in [m]. \tag{17}$$

The proofs appear in Appendix C.4. These results suggest that the bias of gradient-descent based robust ERM over diagonal networks is towards minimum $\ell_1$ solutions, which as we argued in the previous section can have very different robust error compared to $\ell_2$ solutions, which are returned by gradient-descent based robust ERM over linear models. We verify this in the simulations of Section 4.1.

*Remark* 3.9. Technically, Corollary 3.7 only proves convergence to a first order (KKT) point, so we cannot conclude equivalence with the minimum of the $\ell_1$ problem in eq. 17. Yet, we believe that global optimality, under the condition of $(\epsilon, p)$ separability, can be proven by extending the techniques of [Moroshko et al., 2020] in robust ERM.

# 4 Experiments

In this section, we explore with simulations how the implicit bias of optimization in robust ERM is affecting the (robust) generalization of the models. Appendix F contains full experimental details.

## 4.1 Linear models

**Setup** We compare different steepest descent methods in minimizing a worst-case loss with either linear models or diagonal neural networks on synthetic data, and study their robust generalization error. In accordance with Section 2.2, we consider distributions that come from a "teacher" $\mathbf{w}^\star$ with $y = \text{sgn}(\langle \mathbf{w}^\star, \mathbf{x} \rangle)$ that can have *sparse* or *dense* $\mathbf{x}, \mathbf{w}^\star$. We denote by $k_{\mathcal{W}}$ and $k_{\mathcal{X}}$ the expected number of non-zero entries of the ground truth $\mathbf{w}^\star$ and the samples $\mathbf{x}$, respectively. We train linear models $f_{\text{lin}}(\mathbf{x}; \mathbf{w}) = \langle \mathbf{w}, \mathbf{x} \rangle$ with steepest descent with respect to either the $\ell_1$ (coordinate descent - `CD`) or the $\ell_2$ norm (gradient descent - `GD`), and diagonal neural networks $f_{\text{diag}}(\mathbf{x}; \mathbf{u}_+, \mathbf{u}_+) = \langle \mathbf{u}_+^2 - \mathbf{u}_-^2, \mathbf{x} \rangle$ with gradient descent (`diag-net-GD`). We consider $\ell_\infty$ perturbations. We design the following experiment: first, we fit the training data with `CD` for $\epsilon = 0$ and we obtain the value of the $\ell_\infty$ margin of the dataset (denoted as $\epsilon^\star$) at the end of training[2]. This supplies us with an upper bound on the value of $\epsilon$ for our robust ERM experiments, i.e. we know there exists a linear model with $100\%$ robust train accuracy for $\epsilon$ less than or equal to this $\epsilon^\star$ margin. We then perform robust ERM with (full batch) `GD`/`CD`/`diag-net-GD` for various values of $\epsilon$ less than $\epsilon^\star$. We repeat the above for multiple values of dataset size $m$ (and draws of the dataset), and aggregate the results.

**Results** We plot the (robust) generalization gap of the three learning algorithms versus the dataset size for $(k_{\mathcal{W}}, k_{\mathcal{X}})$=(512, 512) (*Dense, Dense*) and $(k_{\mathcal{W}}, k_{\mathcal{X}})$=(4, 512) (*4-Sparse, Dense*) in Figures 1 (bottom) and 2 (left), respectively. In each figure, we show the performance of the methods both in ERM (no perturbations during training) and robust ERM. The evaluation is w.r.t. the $\epsilon$ used in training. For both distributions, we observe a significant change in the relative performance of the methods, when we pass from ERM to robust ERM. For data with a sparse teacher (Figure 2), `CD` and `diag-net-GD` already outperform `GD` in terms of generalization when implementing ERM, as a result of their bias towards minimum $\ell_1$ (*sparse*) solutions. However, in agreement with the bounds of Section 2.2, the interval between the algorithms grows when performing robust ERM as a result of their different biases. In the case of *Dense, Dense* data (Figure 1), the effect of robust ERM is more dramatic, as the algorithms generalize similarly when implementing ERM, yet their gap between their robust generalization in robust ERM exceeds $20\%$ in the case of few training data! Notice that the bounds in Section 2.2 were less optimistic than the experiments show for the performance of `CD` and `diag-net-GD` in this case. Plots with other distributions appear in Figure 5.

To get a fine-grained understanding of the interactions between the hyperparameters of the learning problem, we measure the *average difference* of (robust) generalization gaps between `GD` and `CD`. In particular, for each different combination of sparsities $(k_{\mathcal{W}}, k_{\mathcal{X}})$ and perturbation $\epsilon$, we summarize curves

---

[2]Running this algorithm to convergence is guaranteed to result in the largest possible $\ell_\infty$ separator of the training data [Gunasekar et al., 2018a]. Recall that the $\ell_\infty$-margin is $\max_{\mathbf{w}\neq 0} \min_{i\in[m]} y_i \frac{\langle \mathbf{x}_i, \mathbf{w} \rangle}{\|\mathbf{w}\|_1}$.

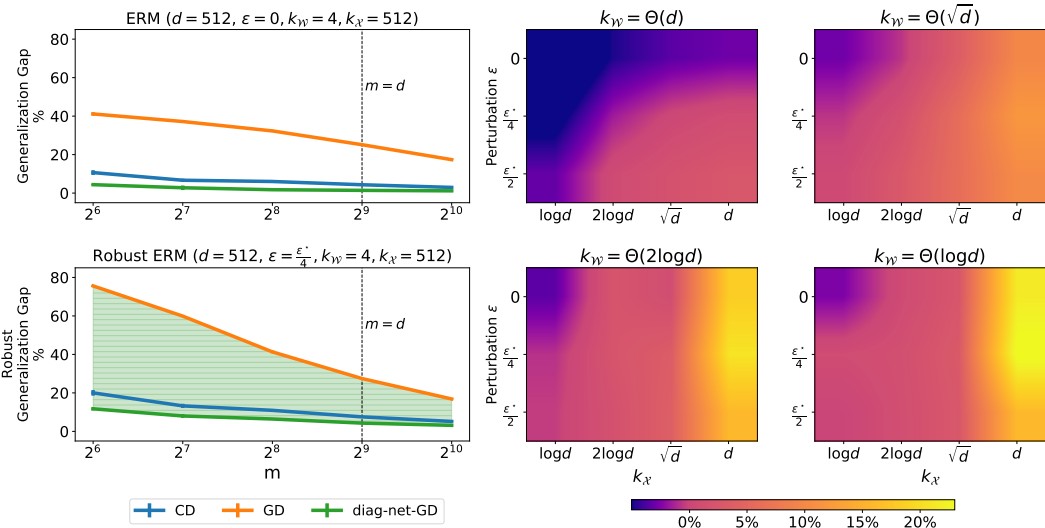

Figure 2: **Left:** Binary classification of data coming from a sparse teacher $\mathbf{w}^\star$ and dense $\mathbf{x}$, with (*bottom*) or without (*top*) $\ell_\infty$ perturbations of the input in $\mathbb{R}^d$ using linear models. We plot the (robust) generalization gap, i.e., (robust) train minus (robust) test accuracy, of different learning algorithms versus the training size $m$. For robust ERM, $\epsilon$ is set to be $\frac{1}{4}$ of the largest permissible value $\epsilon^\star$. The gap between the methods grows when we pass from ERM to robust ERM. **Right:** Average benefit of CD over GD (in terms of generalization gap) for different values of teacher sparsity $k_\mathcal{W}$, data sparsity $k_\mathcal{X}$ and magnitude of $\ell_\infty$ perturbation $\epsilon$.

of the form of Figure 2 (left) into one number, by calculating: $\frac{1}{2^{10}-2^6} \int_{2^6}^{2^{10}} (\mathtt{GD}(m) - \mathtt{CD}(m))\,dm$. The results are shown in Figure 2 (right). Notice that, as argued in Section 2.2, there are cases with $\epsilon > 0$ where CD does not outperform GD ($k_\mathcal{W} = \Theta(d)$, $k_\mathcal{X} = \Theta(\log d)$), because the learning problem is much more "skewed" towards dense solutions. We also observe that when $\epsilon$ goes from 0 to $\frac{\epsilon^\star}{4}$ the edge of CD over GD grows. Past a certain threshold of $\epsilon$, the two methods will start to perform the same, since for $\epsilon = \epsilon^\star$ the algorithms return the same solution (Lemma C.7). See also Appendix E and Figure 6 for the average difference of "clean" generalization gaps between GD and CD.

## 4.2 Neural networks

Our discussion has focused so far on linear (with respect to the input) models, where a closed form solution for the worst-case loss allowed us to obtain precise answers for the connection between generalization and optimization bias in robust ERM. Such a characterization for general models is too optimistic at this point, because, even for a kernelized model $f(\mathbf{x}; \mathbf{w}) = \langle \mathbf{w}, \phi(\mathbf{x}) \rangle$, it is not clear how to compute the right notion of margin that arises from $\min_{\|\mathbf{x}' - \mathbf{x}\|_p \le \epsilon} \langle \mathbf{w}, \phi(\mathbf{x}') \rangle$ without making further assumptions about $\phi(\cdot)$. As such, it is difficult to reason that one set of optimization choices will lead to better suited implicit bias than another. We assess, however, experimentally, what effect (if any) the choice of the optimization algorithm has on the robustness of a non-linear model. To this end, we train neural networks with two optimization algorithms, gradient descent (GD) and sign (gradient) descent (SD) for various values of perturbation magnitude $\epsilon$, focusing on $\ell_\infty$ perturbations. SD corresponds to steepest descent with respect to the $\ell_\infty$ norm and is expected to obtain a minimum with very different properties than the one obtained with GD (Appendix D). In practice, we found it easier to train neural networks with SD than with any other steepest descent algorithm (besides GD).

**Fully Connected NNs**   We first focus on ReLU networks with 1 hidden layer without a bias term: $f(\mathbf{x}) = \sum_{j=1}^{k} \mathbf{u}_j \sigma(\mathbf{W}_j \mathbf{x})$, where $\sigma(u) = \max(0, u)$ is applied elementwise. For this class of homogeneous networks, we expect very different implicit biases when performing (robust) ERM with GD versus SD (see Appendix D for details). In Figure 3, we plot the accuracy of models trained on random subsets of MNIST [LeCun et al., 1998] with "standard" ERM ($\epsilon = 0$) and robust ERM ($\epsilon = 0.2$). We observe that in ERM (*top*), the choice of the algorithm does not affect the generalization

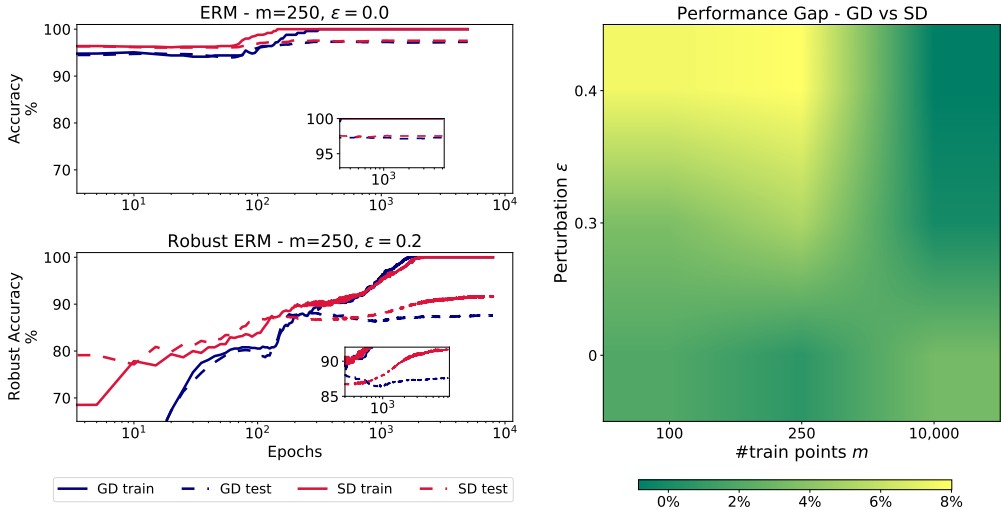

Figure 3: **Left:** Comparison of two optimization algorithms, gradient descent and sign gradient descent, in ERM and robust ERM on a subset of MNIST (digits 2 vs 7) with 1 hidden layer ReLU nets. Train and test accuracy correspond to the magnitude of perturbation $\epsilon$ used during training. We observe that in robust ERM the gap between the generalization of the two algorithms increases. **Right:** Gap in (robust) test accuracy (with respect to the $\epsilon$ used in training) of CNNs trained with `GD` and `SD` (`GD` accuracy minus `SD` accuracy) on subsets of MNIST (all classes) for various of $\epsilon$ and $m$.

error much. But, for $\epsilon = 0.2$ (*bottom*), `SD` significantly outperforms `GD` (4.11% mean difference over 3 random seeds), even though both algorithms reach 100% robust train accuracy. Notably, in this case, robust ERM with `SD` not only achieves smaller robust generalization error, but also avoids robust overfitting during training, in contrast to `GD`. It is plausible that robust overfitting, which gets observed during the late phase of training [Rice et al., 2020], is due to (or attenuated by) the implicit bias of an algorithm kicking in late during robust ERM. This bias can either aid or harm the robust generalization of the model and perhaps this is why the two algorithms exhibit different behavior. It would be interesting for future work to further study this connection. See Appendix E for plots with different values of $\epsilon$ and $m$.

**Convolutional NNs**  Departing from the homogeneous setting, where the implicit bias of robust ERM is known or can be "guessed", we now train convolutional neural networks (with bias terms). As a result, we do not have direct control over which biases our optimization choices will elicit, but changing the optimization algorithm should still yield biases towards minima with different properties. In Figure 3, we plot the mean difference (over 3 random seeds) between the generalization of the converged models. We see that the harder the problem is (fewer samples $m$, request for larger robustness $\epsilon$), the bigger the price of implicit bias becomes. Note that for this architecture it turns out that the implicit bias of `GD` is better "aligned" with our learning problem and `GD` generalizes better than `SD`, despite facing the opposite situation in homogeneous networks. This should not be entirely surprising, since we saw already in linear models that a reparameterization can drastically change the induced bias of the same algorithm.

## 5   Conclusion

In this work, we studied from the perspective of learning theory the issue of the large generalization gap when training robust models and identified the implicit bias of optimization as a contributing factor. Our findings seem to suggest that optimizing models for robust generalization is challenging because it is tricky to do *capacity control* "right" in robust machine learning. The experiments of Section 4 seem to suggest searching for different first-order optimization algorithms (besides gradient descent) for robust ERM (adversarial training) as a promising avenue for future work.

**Acknowledgments.** NT and JK acknowledge support through the NSF under award 1922658. Supported in part by the NSF-Simons Funded Collaboration on the Mathematics of Deep Learning (https://deepfoundations.ai/), the NSF TRIPOD Institute on Data Economics Algorithms and Learning (IDEAL) and an NSF-IIS award. Part of this work was done while NT was visiting the Toyota Technological Institute of Chicago (TTIC) during the winter of 2024, and NT would like to thank everyone at TTIC for their hospitality, which enabled this work. This work was supported in part through the NYU IT High Performance Computing resources, services, and staff expertise.

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

# A  Related work

**Most relevant to our work**  Yin et al. [2019], Awasthi et al. [2020] derived generalization bounds for adversarially robust classification for linear models and simple neural networks, based on the notion of Rademacher Complexity [Koltchinskii and Panchenko, 2002], and form the starting point of our work (Section 2). Gunasekar et al. [2018a] studied the implicit bias of steepest descent in ERM for linear models, while Li et al. [2020] analyzed the implicit bias of gradient descent in robust ERM. Theorem 3.3 can be seen as a generalization of these results. In Section 3.2, we analyze robust ERM with gradient descent in diagonal neural networks, which were introduced in [Woodworth et al., 2020] as a model of feature learning in deep neural networks. The work of Faghri et al. [2021] discusses the connection between optimization bias and adversarial robustness, yet with a very different focus than ours; the authors identify conditions where "standard" ERM produces maximally robust classifiers, and, leveraging results on the implicit bias of CNNs [Gunasekar et al., 2018b], they design a new adversarial attack that operates in the frequency domain.

**Adversarial Robustness**  Our discussion is focused on so-called white-box robustness (where an adversary has access to any information about the model) - see [Papernot et al., 2016] for a taxonomy of threat models. Adversarial examples in machine learning were first studied in [Biggio et al., 2013] for simple models and in [Szegedy et al., 2014] for deep neural networks deployed in image classification tasks. The adversarial vulnerability of neural networks is ubiquitous [Papernot et al., 2016] and it has been observed in many settings and several modalities (see, for instance, [Kurakin et al., 2017a, Jia and Liang, 2017, Zou et al., 2023]). The reasons for that still remain unclear - explanations in the past have entertained hypotheses such as high dimensionality of input space [Fawzi et al., 2018, Gilmer et al., 2018, Shafahi et al., 2019a], presence of spurious features in natural data [Ilyas et al., 2019, Tsipras et al., 2019, Tsilivis and Kempe, 2022], limited model complexity [Nakkiran, 2019], fundamental computational limits of learning algorithms [Bubeck et al., 2019], and the implicit bias of standard (non-robust) algorithms [Frei et al., 2023, Vardi et al., 2022]. Many empirical methods for defending neural networks have been proposed, but most of them failed to conclusively solve the issue [Carlini and Wagner, 2017, Athalye et al., 2018]. The only mechanism that can be adapted to any threat model and has passed the test of time is Adversarial Training [Madry et al., 2018, Goodfellow et al., 2015, Kurakin et al., 2017b, Shaham et al., 2018], i.e., robust ERM. For neural network training, this translates to calculating at each step adversarial examples with (projected) gradient ascent (or some variant), and then updating the weights with gradient descent (using the gradient of the loss evaluated on the adversarial points). There have been many attempts on improving this method, either computationally by reducing the amount of gradient calculations [Shafahi et al., 2019b, Zhang et al., 2019a, Wong et al., 2020], or statistically by modifying the loss function [Zhang et al., 2019b, Awasthi et al., 2023]. A common pitfall of all these methods is large (robust) generalization gap [Croce et al., 2021] and (robust) overfitting during training [Rice et al., 2020]; towards the end of training, the robust test error increases even though the robust training error continues to decrease. Vast amounts of synthetic training data have been shown to help on both accounts, alleviating the need for early stopping during training [Wang et al., 2023].

**Margin-based Generalization bounds**  The idea of (confidence) margin has been central in the development of many machine learning methods [Vapnik, 1998, Taskar et al., 2003, Rudin et al., 2005] and it has been used in several contexts for justifying their empirical success [Cortes and Vapnik, 1995, Schapire et al., 1997, Koltchinskii and Panchenko, 2002]. In linear models and kernel methods, it is closely related to the notion of geometric margin, and margin-based generalization bounds can explain the strong generalization performance in high-dimensions [Vapnik, 1998, Mohri et al., 2012, Shalev-Shwartz and Ben-David, 2014]. For neural networks, they are still the object of active research [Bartlett et al., 2017, Neyshabur et al., 2018, Long and Sedghi, 2020, Cortes et al., 2021]. For these kind of bounds, the Rademacher complexity of the hypothesis class plays a central role [Koltchinskii, 2001]. Rademacher complexity-type analyses have been shown to subsume other similar frameworks [Kakade et al., 2008, Foster et al., 2019], such as the PAC-Bayes one [McAllester, 1998], and in many cases they can provide the finest known guarantees. Yin et al. [2019] and Awasthi et al. [2020] recently derived margin-based bounds for adversarially robust classification - see also Mustafa et al. [2022] for non-additive perturbations.

**Implicit Bias of Optimization Algorithms**  The implicit bias (or regularization) of optimization algorithms refers to the tendency of gradient methods to induce properties to the solution that were

not explicitly specified. It is believed to be beneficial for generalization in learning [Neyshabur et al., 2015]. The implicit bias of gradient descent towards margin maximization/norm minimization has been studied in many learning setups including matrix factorization [Arora et al., 2019, Gunasekar et al., 2017], learning with linear models [Soudry et al., 2018, Ji and Telgarsky, 2019b], deep linear [Ji and Telgarsky, 2019a] and convolutional networks [Gunasekar et al., 2018b], and homogeneous models [Lyu and Li, 2020, Nacson et al., 2019]. Telgarsky [2013] and Gunasekar et al. [2018a] have analyzed implicit biases beyond $\ell_2$-like margin maximization for other optimization algorithms; namely Adaboost and Steepest (and Mirror) Descent, respectively. See Vardi [2023] for a comprehensive survey of the area. The importance of sparsity (min-$\ell_1$ solutions) in binary classification has been studied in [Ng, 2004]. In the context of adversarial training, Li et al. [2020] analyzed the implicit bias of optimizing a worst-case loss with gradient descent in linear models, and Lyu and Zhu [2022] extended these results to deep models.

# B  Generalization Bounds for Robust Interpolators

In this Section, we provide the proof of Proposition 2.2, which we now restate for convenience.

**Proposition B.1.** *(Generalization bound for robust interpolators) Consider a distribution $\mathcal{D}$ over $\mathbb{R}^d \times \{\pm 1\}$ with $\mathbb{P}_{(\mathbf{x},y)\sim\mathcal{D}} \left[ y = \operatorname{sgn}(\langle \mathbf{w}^\star, \mathbf{x} \rangle),\ \forall \mathbf{x}' : \|\mathbf{x}' - \mathbf{x}\|_p \leq \epsilon \right] = 1$ for some $\mathbf{w}^\star \in \mathbb{R}^d$. Let $S \sim \mathcal{D}^m$ be a draw of a random dataset $S = \{(\mathbf{x}_1, y_1), \ldots, (\mathbf{x}_m, y_m)\}$ and let $\mathcal{H}'_r = \left\{ \mathbf{x} \mapsto \langle \mathbf{w}, \mathbf{x} \rangle : \|\mathbf{w}\|_r \leq \|\mathbf{w}^\star\|_r \ \wedge \ \mathbf{w} \in \operatorname{argmax}_{\|\mathbf{u}\|_r \leq 1} \min_{i \in [m]} \min_{\|\mathbf{x}'_i - \mathbf{x}\|_p \leq \epsilon} y_i \langle \mathbf{u}, \mathbf{x}'_i \rangle \right\}$ be a hypothesis class of maximizers of the robust margin. Then, for any $\delta > 0$, with probability at least $1 - \delta$ over the draw of the random dataset $S$, for all $h \in \mathcal{H}'_r$, it holds:*

$$\widetilde{L}_{\mathcal{D},01}(h) \leq \begin{cases} \frac{2}{\sqrt{m}} \left( \max_i \|\mathbf{x}_i\|_\infty \|\mathbf{w}^\star\|_1 \sqrt{2 \log(2d)} + \epsilon \|\mathbf{w}^\star\|_1 \right) + 3\sqrt{\frac{\log 2/\delta}{2m}}, & r = 1 \\ \frac{2}{\sqrt{m}} \left( \max_i \|\mathbf{x}_i\|_2 \|\mathbf{w}^\star\|_2 + \epsilon \|\mathbf{w}^\star\|_2 d^{\max\left(\frac{1}{p^\star} - \frac{1}{2}, 0\right)} \right) + 3\sqrt{\frac{\log 2/\delta}{2m}}, & r = 2. \end{cases} \tag{18}$$

*Proof.* First, notice that $\mathcal{H}'_r \subseteq \mathcal{H}_r$, so, by the definition of the Rademacher complexity, it holds: $\hat{\mathfrak{R}}_S(\mathcal{H}'_r) \leq \hat{\mathfrak{R}}_S(\mathcal{H}_r)$. Thus, from Theorem 2.1, we have for all $h \in \mathcal{H}'_r$ and for $\rho > 0$ with probability $1 - \delta$:

$$\widetilde{L}_{\mathcal{D}}(h) \leq \widetilde{L}_S(h) + \frac{2}{\rho} \hat{\mathfrak{R}}_S(\widetilde{\mathcal{H}}_r) + 3\sqrt{\frac{\log 2/\delta}{2m}}. \tag{19}$$

Observe that the ramp loss $l_\rho(u) = \min(1, \max(0, 1 - \frac{u}{\rho})), \rho > 0$ is an upper bound on the 0-1 loss, thus we readily get a bound for the 0-1 robust risk:

$$\widetilde{L}_{\mathcal{D},01}(h) \leq \widetilde{L}_S(h) + \frac{2}{\rho} \hat{\mathfrak{R}}_S(\widetilde{\mathcal{H}}_r) + 3\sqrt{\frac{\log 2/\delta}{2m}}. \tag{20}$$

Now, let us specialize the ramp loss for $\rho = 1$ (a stronger version of this Proposition can be obtained for a $\rho$ that depends on the data - see, for instance, the techniques in Theorem 5.9 in [Mohri et al., 2012]). Then, the bound becomes:

$$\widetilde{L}_{\mathcal{D},01}(h) \leq \frac{1}{m} \sum_{i=1}^{m} \max_{\|\mathbf{x}'_i - \mathbf{x}_i\|_p \leq \epsilon} \min(1, \max(0, 1 - y_i \langle \mathbf{w}, \mathbf{x}'_i \rangle)) + 2\hat{\mathfrak{R}}_S(\widetilde{\mathcal{H}}_r) + 3\sqrt{\frac{\log 2/\delta}{2m}}. \tag{21}$$

But, notice that for all $h \in \mathcal{H}'_r$ and their corresponding $\mathbf{w}$, the empirical loss $\frac{1}{m} \sum_{i=1}^{m} \max_{\|\mathbf{x}'_i - \mathbf{x}_i\|_p \leq \epsilon} \min(1, \max(0, 1 - y_i \langle \mathbf{w}, \mathbf{x}'_i \rangle))$ is 0, since:

$$\operatorname*{argmax}_{\substack{\mathbf{w}: \|\mathbf{w}\|_r \leq 1}} \min_{i \in [m]} \min_{\|\mathbf{x}'_i - \mathbf{x}_i\|_p \leq \epsilon} y_i \langle \mathbf{w}, \mathbf{x}'_i \rangle = \operatorname*{argmax}_{\substack{\mathbf{w} \in \mathbb{R}^d \\ \mathbf{w} \neq \mathbf{0}}} \min_{i \in [m]} \min_{\|\mathbf{x}'_i - \mathbf{x}_i\|_p \leq \epsilon} \frac{y_i \langle \mathbf{w}, \mathbf{x}'_i \rangle}{\|\mathbf{w}\|_r}$$

$$= \operatorname*{argmax}_{\substack{\mathbf{w} \in \mathbb{R}^d, \mathbf{w} \neq \mathbf{0}: \\ \min_{i \in [m]} \min_{\|\mathbf{x}'_i - \mathbf{x}_i\|_p \leq \epsilon} y_i \langle \mathbf{w}, \mathbf{x}'_i \rangle = 1}} \frac{1}{\|\mathbf{w}\|_r} \tag{22}$$

$$= \operatorname*{argmin}_{\substack{\mathbf{w} \in \mathbb{R}^d, \mathbf{w} \neq \mathbf{0}: \\ \min_{\|\mathbf{x}'_i - \mathbf{x}_i\|_p \leq \epsilon} y_i \langle \mathbf{w}, \mathbf{x}'_i \rangle \geq 1\ \forall i \in [m]}} \|\mathbf{w}\|_r.$$

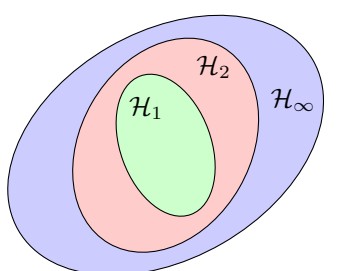

Figure 4: An illustration of the model selection problem we are facing in Section 2. We depict hypothesis classes which correspond to $\mathcal{H}_r = \{\mathbf{x} \mapsto \langle \mathbf{w}, \mathbf{x} \rangle : \|\mathbf{w}\|_r \leq \mathcal{W}\}$ for $r = 1, 2, \infty$ (notice that here, for illustration purposes, we keep $\mathcal{W}$ constant and not dependent on $r$). Increasing the order $r$ of $\mathcal{H}_r$ can decrease the approximation error of the class, but it might increase the complexity captured by the worst-case Rademacher Complexity term of eq. (6).

Note that the set of solutions is not empty, since $\mathbf{w}^\star$ satisfies the constraints with probability 1. As a result, for all $h \in \mathcal{H}'_r$ we obtain:

$$\widetilde{L}_{\mathcal{D},01}(h) \leq 2\hat{\mathfrak{R}}_S(\widetilde{\mathcal{H}}_r) + 3\sqrt{\frac{\log 2/\delta}{2m}}. \tag{23}$$

Combining this with the upper bound of the adversarial Rademacher complexity of eq. (6), together with the standard Rademacher complexity bounds for $r = 1, 2$ [Kakade et al., 2008]:

$$\hat{\mathfrak{R}}_S(\mathcal{H}_r) \leq \begin{cases} \mathcal{O}\left(\frac{\max_{i \in [m]} \|\mathbf{x}_i\|_\infty \mathcal{W}_1 \sqrt{2\log 2d}}{\sqrt{m}}\right), \ r = 1, \\ \mathcal{O}\left(\frac{\max_{i \in [m]} \|\mathbf{x}_i\|_2 \mathcal{W}_2}{\sqrt{m}}\right), \ r = 2. \end{cases} \tag{24}$$

we obtain the result. □

| w \ x | Sparse | Dense |
|---|---|---|
| Sparse | $\ell_1, \ell_2$ similar as $\epsilon \to 0$, $\ell_1$ better as $\epsilon \uparrow 0$ | $\ell_1$ better as $\epsilon \to 0, \epsilon \uparrow 0$ |
| Dense | $\ell_2$ better as $\epsilon \to 0, \epsilon \uparrow 0$ | $\ell_1, \ell_2$ similar as $\epsilon \to 0, \epsilon \uparrow 0$ |

Table 1: A summary of the expected generalization behavior for the various distributions of Section 2.2. $\epsilon$ denotes the strength of $\ell_\infty$ perturbations and $\ell_1, \ell_2$ denote the type of regularization applied to the solution.

## C  Implicit Biases in Robust ERM

### C.1  Robust ERM over Linear Models with Steepest Flow

First, we provide the proof of Theorem 3.3. Recall that we are interested in analyzing the implicit bias of steepest descent algorithms with infinitesimal step size when minimizing a worst case loss.

The steepest flow update with respect to a norm $\|\cdot\|$ is written as follows:

$$\frac{d\mathbf{w}}{dt} \in \left\{ \mathbf{v} \in \mathbb{R}^d : \mathbf{v} \in \operatorname*{argmin}_{\mathbf{u} \in \mathbb{R}^d : \|\mathbf{u}\| \leq \|\mathbf{g}\|_\star} \langle \mathbf{u}, \mathbf{g} \rangle, \mathbf{g} \in \partial \widetilde{L}_S \right\} := \mathcal{S}, \tag{25}$$

where recall $\partial \widetilde{L}_S$ is the set of subgradients of $\widetilde{L}_S$.

We restate Theorem 3.3.

**Theorem C.1.** *For any $(\epsilon, p)$-linearly separable dataset and any initialization $\mathbf{w}_0$, steepest flow with respect to the $\ell_r$ norm, $r \geq 1$, on the worst-case exponential loss $\widetilde{L}_S(\mathbf{w}) = \sum_{i=1}^m \max_{\|\mathbf{x}_i' - \mathbf{x}_i\|_p \leq \epsilon} \exp(-y_i \langle \mathbf{w}, \mathbf{x}_i' \rangle)$ satisfies:*

$$\lim_{t \to \infty} \min_i \min_{\|\mathbf{x}_i' - \mathbf{x}_i\|_p \leq \epsilon} \frac{y_i \langle \mathbf{w}_t, \mathbf{x}_i' \rangle}{\|\mathbf{w}_t\|_r} = \max_{\mathbf{w} \neq 0} \min_i \min_{\|\mathbf{x}_i' - \mathbf{x}_i\|_p \leq \epsilon} \frac{y_i \langle \mathbf{w}, \mathbf{x}_i' \rangle}{\|\mathbf{w}\|_r}. \tag{26}$$

*Proof.* Throughout the proof, we suppress the dependence of $\mathbf{w}$ on time. Let us define the maximum, worst-case, margin:

$$\widetilde{\gamma}_{r^\star} = \max_{\mathbf{w} \neq 0} \min_{i \in [m]} \min_{\|\mathbf{x}_i' - \mathbf{x}_i\|_p \leq \epsilon} \frac{y_i \langle \mathbf{w}, \mathbf{x}_i' \rangle}{\|\mathbf{w}\|_r}. \tag{27}$$

We use the subscript $r^\star$, because $\widetilde{\gamma}_{r^\star}$ maximizes distance with respect to the $\ell_{r^\star}$ norm. Recall that the loss function is given as:

$$\widetilde{L}_S(\mathbf{w}) = \sum_{i=1}^m \max_{\|\mathbf{x}_i' - \mathbf{x}_i\|_p \leq \epsilon} \exp(-y_i \langle \mathbf{w}, \mathbf{x}_i' \rangle) = \sum_{i=1}^m \exp(-y_i \langle \mathbf{w}, \mathbf{x}_i \rangle + \epsilon \|\mathbf{w}\|_{p^\star}). \tag{28}$$

By the definition of the loss function, we have for any $t > 0$:

$$\widetilde{L}_S(\mathbf{w}) = \sum_{i=1}^m \exp(-y_i \langle \mathbf{w}, \mathbf{x}_i \rangle + \epsilon \|\mathbf{w}\|_{p^\star}) \geq \max_{i \in [m]} \exp(-y_i \langle \mathbf{w}, \mathbf{x}_i \rangle + \epsilon \|\mathbf{w}\|_{p^\star}). \tag{29}$$

Thus, we obtain the following relation between the loss and the current margin:

$$\min_{i \in [m]} y_i \langle \mathbf{w}, \mathbf{x}_i \rangle - \epsilon \|\mathbf{w}\|_{p^\star} \geq \log \frac{1}{\widetilde{L}_S(\mathbf{w})}, \tag{30}$$

so the goal will be to lower bound the RHS. For that we need the following Lemma, which consists of the core of the proof and quantifies the relation between maximum margin and loss (sub) gradients. This Lemma generalizes Lemma C.1 in [Li et al., 2020] that applies to gradient descent. We will overload notation and denote by $\partial f$ any subgradient of $f$.

**Lemma C.2.** *For any $\mathbf{w} \in \mathbb{R}^d$, it holds:*

$$\widetilde{\gamma}_{r^\star} \leq \frac{\|\partial \widetilde{L}_S(\mathbf{w})\|_{r^\star}}{\widetilde{L}_S(\mathbf{w})}. \tag{31}$$

*Proof.* Let $\widetilde{\mathbf{u}}_{r^\star}$ be a vector that attains $\widetilde{\gamma}_{r^\star}$, i.e. $\widetilde{\mathbf{u}}_{r^\star}$ is a worst-case $\ell_{r^\star}$ maximum margin separator:

$$\widetilde{\mathbf{u}}_{r^\star} \in \operatorname*{argmax}_{\mathbf{w} \neq 0} \min_{i \in [m]} \min_{\mathbf{x}_i' \in \mathcal{B}_\epsilon^{\mathrm{p}}(\mathbf{x})} \frac{y_i \langle \mathbf{w}, \mathbf{x}_i' \rangle}{\|\mathbf{w}\|_r}. \tag{32}$$

Then, we have (since $\widetilde{L}_S$ is convex, the "chain rule" holds):

$$\begin{aligned}
\left\langle \widetilde{\mathbf{u}}_{r^\star}, -\partial \widetilde{L}_S(\mathbf{w}) \right\rangle &= \sum_{i=1}^m \left\langle \widetilde{\mathbf{u}}_{r^\star}, y_i \mathbf{x}_i - \epsilon \partial \|\mathbf{w}\|_{p^\star} \right\rangle e^{-y_i \langle \mathbf{w}, \mathbf{x}_i \rangle + \epsilon \|\mathbf{w}\|_{p^\star}} \\
&= \sum_{i=1}^m \|\widetilde{\mathbf{u}}_{r^\star}\|_r \frac{\langle \widetilde{\mathbf{u}}_{r^\star}, y_i \mathbf{x}_i \rangle - \epsilon \langle \widetilde{\mathbf{u}}_{r^\star}, \partial \|\mathbf{w}\|_{p^\star} \rangle}{\|\widetilde{\mathbf{u}}_{r^\star}\|_r} e^{-y_i \langle \mathbf{w}, \mathbf{x}_i \rangle + \epsilon \|\mathbf{w}\|_{p^\star}}.
\end{aligned} \tag{33}$$

But, by the definition of the dual norm and of the subgradient, we have $\langle \widetilde{\mathbf{u}}_{r^\star}, \partial \|\mathbf{w}\|_{p^\star} \rangle \leq \|\widetilde{\mathbf{u}}_{r^\star}\|_{p^\star} \|\partial \|\mathbf{w}\|_{p^\star}\|_p = \|\widetilde{\mathbf{u}}_{r^\star}\|_{p^\star}$, so eq. (33) becomes:

$$\left\langle \widetilde{\mathbf{u}}_{r^\star}, -\partial \widetilde{L}_S(\mathbf{w}) \right\rangle \geq \sum_{i=1}^m \|\widetilde{\mathbf{u}}_{r^\star}\|_r \widetilde{\gamma}_{r^\star} e^{-y_i \langle \mathbf{w}, \mathbf{x}_i \rangle + \epsilon \|\mathbf{w}\|_{p^\star}}, \tag{34}$$

which by rearranging can be written as:

$$\left\langle \frac{\widetilde{\mathbf{u}}_{r^\star}}{\|\widetilde{\mathbf{u}}_{r^\star}\|_r}, -\partial \widetilde{L}_S(\mathbf{w}) \right\rangle \geq \widetilde{\gamma}_{r^\star} \widetilde{L}_S(\mathbf{w}). \tag{35}$$

Finally, again, by the definition of the dual norm, we get the desired result:

$$\|\partial \widetilde{L}_S(\mathbf{w})\|_{r^\star} \geq \widetilde{\gamma}_{r^\star} \widetilde{L}_S(\mathbf{w}). \tag{36}$$

$\square$

In light of this Lemma, we can lower bound the derivative of the RHS of eq. (30) as follows:

$$
\begin{aligned}
\frac{d\log\frac{1}{\widetilde{L}_S}}{dt} &= -\frac{1}{\widetilde{L}_S}\frac{d\widetilde{L}_S}{dt} \\
&= -\frac{1}{\widetilde{L}_S}\left\langle \partial\widetilde{L}_S, \frac{d\mathbf{w}}{dt}\right\rangle \qquad \text{(Chain rule)} \\
&= \frac{\|\partial\widetilde{L}_S\|_{r^\star}\left\|\frac{d\mathbf{w}}{dt}\right\|_r}{\widetilde{L}_S} \qquad \text{(Def. of steepest flow)} \\
&\geq \widetilde{\gamma}_{r^\star}\left\|\frac{d\mathbf{w}}{dt}\right\|_r \qquad \text{(Lemma C.2).}
\end{aligned}
\tag{37}
$$

Thus, eq. (30) becomes:

$$
\begin{aligned}
\min_{i\in[m]}\frac{y_i\langle\mathbf{w},\mathbf{x}_i\rangle-\epsilon\|\mathbf{w}\|_{p^\star}}{\|\mathbf{w}\|_r} &\geq \widetilde{\gamma}_{r^\star}\frac{\int_0^t\left\|\frac{d\mathbf{w}}{ds}\right\|_r ds}{\|\mathbf{w}\|_r} \qquad \text{(from Eq. (37))} \\
&\geq \widetilde{\gamma}_{r^\star}\frac{\left\|\int_0^t\frac{d\mathbf{w}}{ds}ds\right\|_r}{\|\mathbf{w}\|_r} \\
&= \widetilde{\gamma}_{r^\star}\frac{\|\mathbf{w}-\mathbf{w}_0\|}{\|\mathbf{w}\|_r} \\
&= \widetilde{\gamma}_{r^\star}\left\|\frac{\mathbf{w}}{\|\mathbf{w}\|_r}-\frac{\mathbf{w}_0}{\|\mathbf{w}\|_r}\right\|_r \to \widetilde{\gamma}_{r^\star},
\end{aligned}
\tag{38}
$$

since $\widetilde{L}_S \to 0$ ($\frac{d\widetilde{L}_S}{dt} \leq 0$ - see Lemma C.3 - and $\widetilde{L}_S$ is bounded from below) and hence it must be $\|\mathbf{w}\| \to \infty$. $\qquad\square$

**Lemma C.3.** *For any convex $L$, the steepest flow of eq. (25) satisfies*

$$
\frac{dL}{dt} \leq 0 \text{ and } \frac{d\mathbf{w}}{dt} \in \left\{\underset{\mathbf{u}\in\mathbb{R}^d:\|\mathbf{u}\|\leq\|\mathbf{g}\|_\star}{\operatorname{argmin}}\langle\mathbf{u},\mathbf{g}^\star\rangle : \mathbf{g}^\star \in \underset{\mathbf{g}\in\partial\widetilde{L}_S}{\operatorname{argmin}}\|\mathbf{g}\|\right\} \quad \forall t > 0.
\tag{39}
$$

*Proof.* Since $L$ is convex, the "chain rule holds", that is, for all $\mathbf{g} \in \partial L$ we have:

$$
\frac{dL}{dt} = \left\langle \mathbf{g}, \frac{d\mathbf{w}}{dt}\right\rangle.
\tag{40}
$$

First, apply this to the element of $\partial L$, $\mathbf{g}$, that corresponds to $\frac{d\mathbf{w}}{dt}$, then, by the definition of $\mathcal{S}$ and that of a dual norm, we have:

$$
\frac{dL}{dt} = -\left\|\frac{d\mathbf{w}}{dt}\right\|^2.
\tag{41}
$$

Now, apply the "chain rule" for $\mathbf{g}^\star = \operatorname{argmin}_{\mathbf{g}\in\partial L}\|\mathbf{g}\|_\star$:

$$
\frac{dL}{dt} = \left\langle\mathbf{g}^\star, \frac{d\mathbf{w}}{dt}\right\rangle \geq -\|\mathbf{g}^\star\|_\star\left\|\frac{d\mathbf{w}}{dt}\right\|,
\tag{42}
$$

Equating $\frac{dL}{dt}$ from (41), (42), we get $\left\|\frac{d\mathbf{w}}{dt}\right\| \leq \|\mathbf{g}^\star\|_\star$. $\qquad\square$

## C.2 Existence of Steepest Descent Flows

In this section, we prove that $C^1$ steepest descent flows exist for certain norms when the initialization has sufficiently small robust risk (see Theorem C.6 below). This criterion applies to $r$-norms with $r \in (1,\infty)$, but not $r = 1$ or $r = \infty$.

**Lemma C.4.** *If the norm $\|\cdot\|$ is strictly convex and $\mathbf{x} \neq \mathbf{0}$, then there a unique minimizer to $\mathbf{u} \mapsto \langle\mathbf{u},\mathbf{x}\rangle$ over the ball $\overline{B_M(\mathbf{0})} = \{\mathbf{u} : \|\mathbf{u}\| \leq M\}$ for any $M > 0$.*

*Proof.* We'll show this statement via a proof by contrapositive.

Let $u_1$, $u_2$ be any two minimizers of $\langle \mathbf{u}, \mathbf{x} \rangle$ over $\overline{B_M(\mathbf{0})}$. Then because this function is linear, $\mathbf{u}_1$, $\mathbf{u}_2$ cannot be in the interior of $\overline{B_M(\mathbf{0})}$. Due to the convexity of the ball $\overline{B_M(\mathbf{0})}$, the linear combination $t\mathbf{u}_1 + (1 - t)\mathbf{u}_2$ is not in the interior of the ball $\overline{B_M(\mathbf{0})}$, and thus the norm $\|\cdot\|$ is not strictly convex. $\qquad\square$

This result implies that whenever the norm $\|\cdot\|$ is strictly convex, the function $\mathbf{P} : \mathbb{R}^d - \{\mathbf{0}\} \to \mathbb{R}^d$ defined by

$$\mathbf{P}(\mathbf{x}) = \operatorname*{argmax}_{\|\mathbf{u}\| \leq \|\mathbf{x}\|_*} \langle \mathbf{u}, \mathbf{x} \rangle \tag{43}$$

is well-defined.

This observation together with classical results from ODE theory can be leveraged to proved the existence of well behaved steepest descent flows. See Theorem 1.2 [Coddington and Levinson, 1955, pages 6 and 19] for Peano's existence theorem:

**Theorem C.5** (Peano's existence theorem). *Consider the ODE*

$$\mathbf{x}'(t) = \mathbf{f}(t, \mathbf{x}(t))$$

*with initial condition $\mathbf{x}(\tau) = \boldsymbol{\xi}$. If $\mathbf{f} : \mathbb{R} \times \mathbb{R}^d \to \mathbb{R}^d$ is continuous on a rectangle containing $(\tau, \boldsymbol{\xi})$, then there is a $C^1$ solution for $|t - \tau| \leq \alpha$, for some $\alpha > 0$.*

This result states that if the map $f$ is sufficiently well-behaved, then there is a solution to the ODE for sufficiently small time. Below, we prove that there exists a steepest descent flow for all times $t$ by "stitching together" small time intervals for which there exists local solutions. We also require the initial point to satisfy a certain condition so that we avoid the singularity at zero in the map $\mathbf{P}$.

**Theorem C.6.** *Let $\|\cdot\|$ be a strictly convex norm for which the function defined by (43) is continuous. Then if the initial point $\mathbf{w}_0$ satisfies $\mathcal{L}_S(\mathbf{w}_0) < \mathcal{L}_S(\mathbf{0})$ then there exists a $C^1$ steepest descent flow for the equation*

$$\frac{d\mathbf{w}}{dt} = -\mathbf{P}\left(\nabla \mathcal{L}_S(\mathbf{w}(t))\right).$$

*Proof.* Theorem C.5 proves the existence of a $C^1$ local solution for small times $t$. Now let

$$T = \sup\{s : \mathbf{w}(s) \neq \mathbf{0}, \text{ or there exists a } C^1 \text{ solution for all } t < s\}$$

For contradiction, we will assume that $T < \infty$. First, notice that

$$\frac{d}{dt}\mathcal{L}_S(\mathbf{w}(t)) = -\mathbf{P}\left(\nabla \mathcal{L}_S(\mathbf{w}(t))\right) \cdot \nabla \mathcal{L}_S(\mathbf{w}(t)) = -\|\nabla \mathcal{L}_S(\mathbf{w}(t))\|_*^2$$

Thus $\mathcal{L}_S(\mathbf{w}(T)) \leq \mathcal{L}_S(\mathbf{w}_0)) < \mathcal{L}_S(\mathbf{w}(0))$, and consequently, $\mathbf{w}(T) \neq \mathbf{0}$. Therefore, Peano's existence theorem again implies the existence of a local solution starting from $\mathbf{w}(T)$. Thus the solution can be extended past time $T$, which contradicts the definition of $T$. $\qquad\square$

One can show that if $\|\cdot\|$ is the $r$-norm for $r \in (1, \infty)$, then the corresponding function $\mathbf{P}$ defined by (43) is

$$\mathbf{P}(\mathbf{x}) = \|\mathbf{x}\|_{r^*}^{\frac{r-2}{r-1}} \operatorname{sign}(\mathbf{x})|\mathbf{x}\|^{\frac{1}{r-1}}$$

and this function is continuous in $\mathbf{x}$ on the domain $\mathbb{R}^d - \{\mathbf{0}\}$.

Consequently, there exists a steepest descent flow for the $r$-norm for $r \in (1, \infty)$ so long as the initialization satisfies $\mathcal{L}_S(\mathbf{w}_0) < \mathcal{L}_S(\mathbf{0})$.

### C.3 Equivalence of max-margin solutions

**Lemma C.7.** *Let $\{(\mathbf{x}_i, y_i)\}_{i=1}^m$ be a dataset with $\ell_p$ margin equal to $\epsilon^\star$. Any hyperplane that separates $\left\{\left\{\mathbf{x}_i' \in \mathbb{R}^d : \|\mathbf{x}_i' - \mathbf{x}_i\|_p \leq \epsilon^\star\right\}, y_i\right\}_{i=1}^m$ is an $\ell_r$ max-margin separator for any $r$.*

This result informs us that any choice of algorithm in robust ERM (w.r.t to $\ell_p$ perturbations) for $\epsilon$ equal to the $\ell_p$ margin of the dataset will produce the same solutions.

We provide the proof of Lemma C.7. First, one can calculate the margin of a point $\mathbf{x}$ with respect to a linear separator $\mathbf{w}$:

**Lemma C.8.** *The $\ell_p$ margin of a linear hyperplane $\mathbf{w}$ at a datapoint $(\mathbf{x}, y)$ is $y\langle \mathbf{w}, \mathbf{x}\rangle / \|\mathbf{w}\|_{p^*}$.*

*Proof.* We want to find the largest $c$ for which $\langle \mathbf{w}, \mathbf{x} + c\mathbf{h}\rangle \geq 0$ for all $\|\mathbf{h}\|_p \leq 1$.

for all $\mathbf{h}$ with $\|\mathbf{h}\|_p \leq 1$ Taking an infimum over $h$ results in

$$\langle \mathbf{w}, \mathbf{x}\rangle - c\|\mathbf{w}\|_{p^*} \geq 0$$

and thus $c = \langle \mathbf{w}, \mathbf{x}\rangle / \|\mathbf{w}\|_{p^*}$. □

Next, this lemma allows one to calculate the margin of a ball around a point. We denote by $\mathcal{B}_\epsilon^p(\mathrm{x})$ the $\ell_p$ ball around $\mathbf{x}$, i.e. $\mathcal{B}_\epsilon^p(\mathrm{x}) = \{\mathbf{x}' : \|\mathbf{x}' - \mathbf{x}\|_p \leq \epsilon\}$.

**Lemma C.9.** *The $\ell_r$-margin of the set $B_\epsilon^p(\mathbf{x})$ with label $y$ is*

$$\frac{y\mathbf{w} \cdot \mathbf{x} - \epsilon\|\mathbf{w}\|_{p^*}}{\|\mathbf{w}\|_{r^*}}$$

*Proof.* We want to find the largest constant $c$ for which

$$y(\mathbf{w} \cdot (\mathbf{x} + \mathbf{h}_1 + \mathbf{h}_2)) \geq 0$$

for all $\mathbf{h}_1 \in B_\epsilon^p(\mathbf{0})$ and $\mathbf{h}_2 \in B_c^r(\mathbf{0})$. Taking an infimum over all possible $\mathbf{h}_1$ and $\mathbf{h}_2$ results in

$$y\mathbf{w} \cdot \mathbf{x} - \epsilon\|\mathbf{w}\|_{p^*} - c\|\mathbf{w}\|_{r^*} \geq 0$$

Therefore, the largest such possible $c$ is

$$c = \frac{y\mathbf{w} \cdot \mathbf{x} - \epsilon\|\mathbf{w}\|_{r^*}}{\|\mathbf{w}\|_{r^*}}$$

□

This result immediately implies Lemma C.7:

*Proof of Lemma C.7.* Let $\mathbf{w}^*$ be the $\ell_r$ max-margin hyperplane separating the $\{(B_\epsilon^p(\mathbf{x}_i), y_i)\}_{i=1}^m$. If the $\ell_p$-margin of the dataset is $\epsilon$, then Lemma C.8 implies that

$$\min_{i \in [1,m]} \frac{y\mathbf{w}^* \cdot \mathbf{x} - \epsilon\|\mathbf{w}\|_{p^*}}{\|\mathbf{w}\|_{r^*}} = 0$$

and therefore Lemma C.9 implies that the $\ell_r$ max-margin hyperplane has margin 0. On the other hand, any separating hyperplane for $\{(B_\epsilon^p(\mathbf{x}_i), y_i)\}_{i=1}^m$ has a separation margin that is at worst zero. Therefore, any separating hyperplane is an $\ell_r$ max-margin hyperplane.

□

### C.4 Robust ERM over Diagonal Networks

We first show Corollary 3.7.

*Proof.* A diagonal neural network $f_{\mathrm{diag}}(\mathbf{x}; \mathbf{u})$ is 2-homogeneous, since for any $c > 0$, it holds:

$$f_{\mathrm{diag}}(\mathbf{x}; c\mathbf{u}) = \langle (c\mathbf{u}_+)^2 - (c\mathbf{u}_-)^2, \mathbf{x}\rangle = c^2 \langle \mathbf{u}_+^2 - \mathbf{u}_-^2, \mathbf{x}\rangle. \tag{44}$$

Furthermore, for any $\mathbf{x}$, the optimal perturbation is scale invariant as it is: $\mathrm{argmin}_{\|\delta\|_\infty \leq \epsilon} \langle \mathbf{u}_+^2 - \mathbf{u}_-^2, \mathbf{x} + \delta\rangle = -\epsilon\mathrm{sign}(\mathbf{u}_+^2 - \mathbf{u}_-^2)$, which is scale invariant, i.e., $\mathrm{sign}((\alpha\mathbf{u}_+)^2 - (\alpha\mathbf{u}_-)^2) = \mathrm{sign}(\mathbf{u}_+^2 - \mathbf{u}_-^2)$ for any $\alpha > 0$. Thus, $f_{\mathrm{diag}}$ satisfies the conditions of Theorem 3.6. □

Now, we provide the proof of Proposition 3.8, which states that $\ell_2$ minimization in parameter space is equivalent to $\ell_1$ minimization in predictor space for robust ERM in diagonal networks.

*Proof.* Let us recall the two optimization problems:

$$\min_{\mathbf{u}_+ \in \mathbb{R}^d, \mathbf{u}_- \in \mathbb{R}^d} \frac{1}{2} \left( \|\mathbf{u}_+\|_2^2 + \|\mathbf{u}_-\|_2^2 \right)$$
$$\text{s.t. } \min_{\|\mathbf{x}_i' - \mathbf{x}_i\|_p \leq \epsilon} y_i \left\langle \mathbf{u}_+^2 - \mathbf{u}_-^2, \mathbf{x}_i' \right\rangle \geq 1, \ \forall i \in [m], \tag{45}$$

and

$$\min_{\mathbf{w}} \frac{1}{2} \|\mathbf{w}\|_1$$
$$\text{s.t. } \min_{\|\mathbf{x}_i' - \mathbf{x}_i\|_p \leq \epsilon} y_i \left\langle \mathbf{w}, \mathbf{x}_i' \right\rangle \geq 1, \ \forall i \in [m]. \tag{46}$$

We will show that the two problems share the same optimal value $OPT$. Let $(\widetilde{\mathbf{u}}_+, \widetilde{\mathbf{u}}_-), \mathbf{w}^\star$ be optimal solutions of (45) and (46), respectively, with corresponding values $OPT_A, OPT_B$.

- First, we show that $OPT_B \leq OPT_A$. Let $\hat{\mathbf{w}} = \widetilde{\mathbf{u}}_+^2 - \widetilde{\mathbf{u}}_-^2$, then $\hat{\mathbf{w}}$ satisfy the constraints of (46) and:

$$\|\hat{\mathbf{w}}\|_1 = \|\widetilde{\mathbf{u}}_+^2 - \widetilde{\mathbf{u}}_-^2\|_1 = \sum_{j=1}^d |\widetilde{u}_{+,j} - \widetilde{u}_{-,j}| \, |\widetilde{u}_{+,j} + \widetilde{u}_{-,j}|$$
$$\leq \frac{1}{4} \sum_{j=1}^d \left( \widetilde{u}_{+,j} - \widetilde{u}_{-,j} \right)^2 + \left( \widetilde{u}_{+,j} + \widetilde{u}_{-,j} \right)^2 \tag{47}$$
$$= \frac{1}{2} \left( \|\widetilde{\mathbf{u}}_+\|_2^2 + \|\widetilde{\mathbf{u}}_-\|_2^2 \right) = OPT_A.$$

  As $\hat{\mathbf{w}}$ is a feasible point of (46), it is $OPT_B \leq \|\hat{\mathbf{w}}\|_1$ and we deduce $OPT_B \leq OPT_A$.

- Now, we prove the reverse relation. We decompose $\mathbf{w}^\star$ to its positive and negative part, i.e $\mathbf{w}^\star = \hat{\mathbf{u}}_+^2 - \hat{\mathbf{u}}_-^2$, where, observe, the supports (set of indices with non-zero values) of $\hat{\mathbf{u}}_+, \hat{\mathbf{u}}_-$ do not overlap. Then, $(\hat{\mathbf{u}}_+, \hat{\mathbf{u}}_-)$ satisfy the constraints of (45) and, furthermore:

$$\frac{1}{2} \left( \|\hat{\mathbf{u}}_+\|_2^2 + \|\hat{\mathbf{u}}_-\|_2^2 \right) = \frac{1}{2} \|\mathbf{w}^\star\|_1 \leq OPT_B. \tag{48}$$

  Since $(\hat{\mathbf{u}}_+, \hat{\mathbf{u}}_-)$ is a feasible point of (45), we deduce that $OPT_A \leq OPT_B$.

$\square$

# D Cases of Steepest Descent & Implicit Bias

**Coordinate Descent** In coordinate descent, at each step we only update the coordinate with the largest absolute value of the gradient. Formally, its update is given by:

$$\Delta \mathbf{w}_t \in \text{conv} \left\{ -\frac{\partial \mathcal{L}(\mathbf{w}_t)}{\partial \mathbf{w}_t[i]} \mathbf{e}_i : i = \operatorname*{argmax}_{j \in [d]} \left| \frac{\partial \mathcal{L}(\mathbf{w}_t)}{\partial \mathbf{w}_t[j]} \right| \right\}, \tag{49}$$

where $\mathbf{e}_i, i \in [d]$, denotes the standard basis and $\text{conv}(\cdot)$ stands for the convex hull. Coordinate descent has long been studied for its connection with the $\ell_1$ regularized exponential loss and Adaboost. It corresponds to Steepest Descent with respect to the $\ell_1$ norm, and at each step it holds: $\|\Delta \mathbf{w}\|_1 = \|\nabla \mathcal{L}\|_\infty$. In our experiments, we found it difficult to run robust ERM with coordinate descent for large values of perturbation $\epsilon$, both in linear models and neural networks. Also, it is computationally challenging to scale coordinate descent to large models, since only one coordinate gets updated at a time. These are the main reasons why we chose to experiment with Sign (Gradient) Descent in Section 4.2.

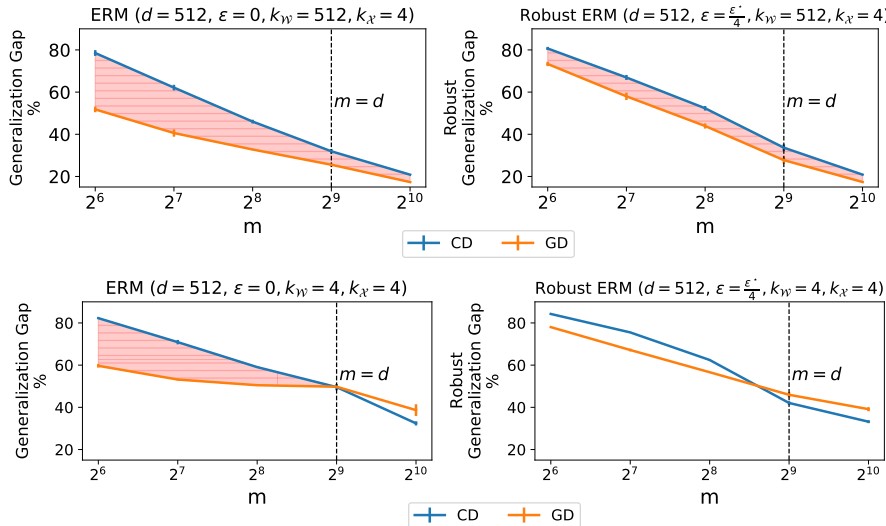

Figure 5: Binary classification of data coming from a dense teacher $\mathbf{w}^\star$ and sparse data $\mathbf{x}$ (top) and from a sparse $\mathbf{w}^\star$ and sparse data $\mathbf{x}$ (bottom). We compare performance of different algorithms with (*right*) or without (*left*) $\ell_\infty$ perturbations of the input in $\mathbb{R}^d$ using linear models. We plot the (robust) generalization gap, i.e., (robust) train minus (robust) test accuracy, of different learning algorithms versus the training size $m$. For robust ERM, $\epsilon$ is set to be $\frac{1}{4}$ of the largest permissible value $\epsilon^\star$. In accordance to the bounds of Section 2.2, it can still be the case that $\ell_2$ solutions will generalize better in robust ERM, due to the significant advantage of them in ERM.

**Sign (Gradient) Descent**    In Sign (Gradient) Descent, we only use the sign of the gradient to update the iterates, i.e.

$$\Delta\mathbf{w}_t = -\text{sign}(\nabla\mathcal{L}(\mathbf{w}_t)). \tag{50}$$

It corresponds to steepest descent with respect to the $\ell_\infty$ norm. Its connection with popular adaptive optimizers has made it an interesting algorithm to study for deep learning applications.

**Implicit bias in homogeneous networks**    From the results of [Lyu and Li, 2020], we know that, for homogeneous networks, gradient descent converges in direction to a KKT point of a maximum margin optimization problem defined by the $\ell_2$ norm. For steepest descent, on the other hand, there is no such characterization, yet we expect a similar result to hold; namely, we expect running ERM with steepest descent to converge in direction to a point that has some relation to the maximum margin optimization problem defined by the norm of the algorithm. By making a leap of faith, we expect something similar to hold for robust ERM. Since the promotion of a margin in one norm can have very different properties from the promotion of a margin in a different norm, we expect robust ERM with gradient descent and sign descent to yield solutions with different properties.

## E   Additional Experiments

**Linear models**    We plot (robust) generalization gaps vs dataset size $m$ for distributions with $(k_{\mathcal{W}}, k_{\mathcal{X}})$ equal to $(512, 4)$ (*Dense, Sparse*) and $(k_{\mathcal{W}}, k_{\mathcal{X}})$ equal to $(4, 4)$ (*Sparse, Sparse*) on the top and the bottom of Figure 5, respectively. In accordance to the bounds of Section 2.2, it can still be the case that $\ell_2$ solutions will generalize better in robust ERM, due to the significant advantage of them in ERM. In Figure 6, we produce heatmaps similar to those of Figure 2, but the benefit of CD over GD is measured with respect to "clean" generalization ($\epsilon = 0$), no matter what value of $\epsilon$ was used during training. In particular, for each combination of data/weight sparsity and perturbation $\epsilon$ used at training, we compute clean generalization gaps of CD and GD solutions for various values of dataset size $m$. We then aggregate the results over $m$ and compute $\frac{1}{2^{10}-2^6}\int_{2^6}^{2^{10}}(\text{GD}(m) - \text{CD}(m))\,dm$, whereas in Section 4.1 the curves $\text{GD}(m), \text{CD}(m)$ referred to the robust error (w.r.t. the value of $\epsilon$ used during training). We observe that there are cases such as the *Dense, Dense* one with $k_{\mathcal{W}} = k_{\mathcal{X}} = d = 512$

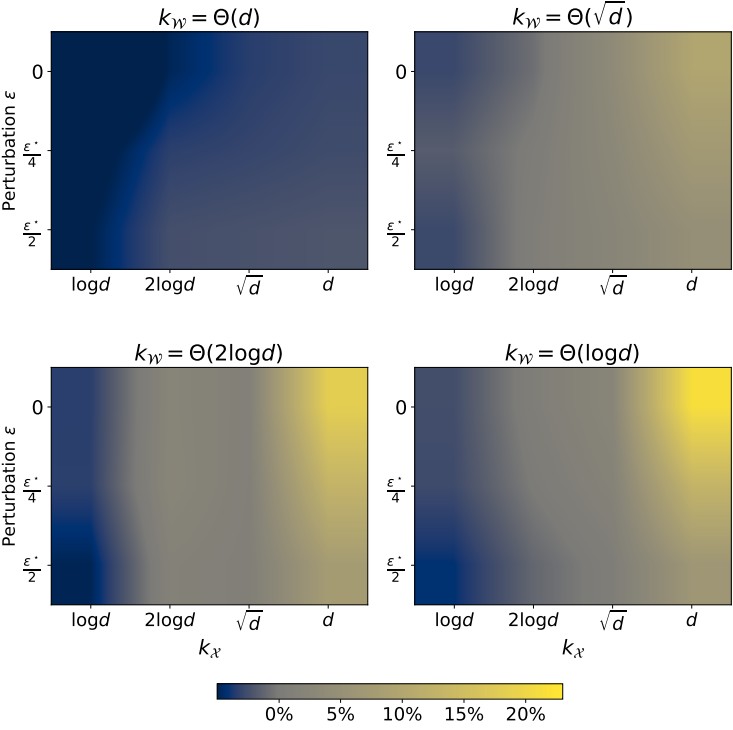

Figure 6: Average benefit, i.e. $\frac{1}{2^{10}-2^6}\int_{2^6}^{2^{10}}\left(\texttt{GD}(m)-\texttt{CD}(m)\right)dm$, of $\texttt{CD}$ over $\texttt{GD}$ (in terms of "clean" generalization gap) for different values of teacher sparsity $k_\mathcal{W}$, data sparsity $k_\mathcal{X}$ and magnitude of perturbation $\epsilon$ used during training (evaluation here is always with respect to $\epsilon = 0$ - "standard" generalization). The dimension $d$ is fixed to be 512.

that for $\epsilon > 0$ (in particular equal to $\epsilon^\star/2$ - bottom right corner in top left subplot), $\texttt{GD}$ generalizes better than $\texttt{CD}$ in terms of clean error, even though it is the other way around for robustness - see Figure 2, Figure 1 (bottom right). This suggests that even if we nail the optimization bias in robust ERM, we might still incur a tradeoff between robustness and accuracy [Tsipras et al., 2019]. However, this need not always be the case; for example for *Sparse, Dense* data, no such tradeoff is observed.

**Fully-Connected Neural Networks**   In Figure 7, we plot (robust) accuracy during training in ERM and robust ERM ($\epsilon = 0.2, 0.3$) for 1 hidden layer ReLU networks trained on a subset of 100 images (randomly drawn in each seed) of digits 2 and 7 from the MNIST dataset. We observe that the gap between the performance of gradient descent and steepest descent in larger in robust ERM than in ERM.

**Convolutional Neural Networks**   In Figure 8, we plot the (robust) train and test accuracy during training for various combinations of dataset size and perturbation magnitude. The main observation, summarized also in Figure 3 (left) in the main text, is that when there are little available data $m$, the implicit bias in robust ERM affects generalization more than in "standard" ERM (row-wise comparison in the Figure). Notice that the artifact of the light-ish bottom right corner in Figure 3 (non-trivial gap between $\texttt{GD}$ and $\texttt{SD}$ for $m = 10,000$ and $\epsilon = 0$) is due to the fact that $\texttt{SD}$ becomes unstable at that time near convergence. Reducing the learning rate would have allievated this "anomaly".

# F   Experimental details

In this Section, we provide more details about our experimental setup. All experiments are implemented in PyTorch and were run either on multiple CPUs (experiments with linear models) or

GPUs. Estimated GPU hours: 200. Link to github repository: `https://github.com/Tsili42/price-imp-bias/tree/main`.

### F.1 Experiments with synthetic data

We consider the following distributions:

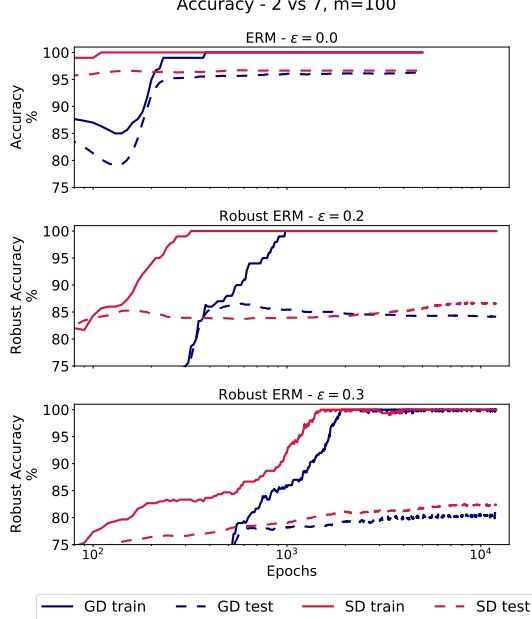

Accuracy - 2 vs 7, m=100

1. *Dense, Dense*: We sample points $\mathbf{x}_i \sim \mathcal{N}(0, I_d), i \in [m]$, and a ground truth vector $\mathbf{w}^\star \sim \mathcal{N}(0, I_d)$ that labels each of the $m$ points with $y_i = \mathrm{sgn}(\langle \mathbf{w}^\star, \mathbf{x}_i \rangle)$.

2. *k-Sparse, Dense*: We sample points $\mathbf{x}_i \sim \{-1, 0, +1\}, i \in [m]$, with corresponding probabilities $\{\frac{k}{2d}, 1 - \frac{k}{d}, \frac{k}{2d}\}$ (so expected number of non-zero entries is $k$) and a ground truth vector $\mathbf{w}^\star \sim \mathcal{N}(0, I_d)$ that labels each of the $m$ points with $y_i = \mathrm{sgn}(\langle \mathbf{w}^\star, \mathbf{x}_i \rangle)$.

3. *Dense, k-Sparse*: Same as before, but now $\mathbf{x}$ is dense and $\mathbf{w}^\star$ is $k$-sparse (with high probability).

4. *k-Sparse, k-Sparse*: We sample points $\mathbf{x}_i \sim \mathcal{N}(0, I_d), i \in [m]$, and a ground truth vector $\mathbf{w}^\star \sim \{-1, 0, +1\}$ with corresponding probabilities $\{\frac{k}{2d}, 1 - \frac{k}{d}, \frac{k}{2d}\}$ (so expected number of non-zero entries is $k$) that labels each of the $m$ points with $y_i = \mathrm{sgn}(\langle \mathbf{w}^\star, \mathbf{x}_i \rangle)$.

Figure 7: Accuracy during training in ERM (*top*) and robust ERM for $\epsilon = 0.2$ (*center*) and $\epsilon = 0.3$ (*bottom*). Setting: 1 hidden layer ReLU networks trained on a subset of MNIST. Mean over 3 random seeds - randomness affects initialization and draw of random dataset. The gap between the generalization of the algorithms is more significant in robust ERM.

**Linear models** For the experiments with linear models $f(\mathbf{x}; \mathbf{w}) = \langle \mathbf{w}, \mathbf{x} \rangle$, we train with the exponential loss and we use an (adaptive) learning rate schedule $\eta_t = \min\{\eta_+, \frac{1}{(B+\epsilon)^2 \widetilde{\mathcal{L}}(w_t)}\}$, where $\eta_+$ is a finite upper bound ($10^5$ in our experiments) and $B$ is the largest $\ell_\infty$ norm of the train data. This type of learning rate schedule can be derived by a discrete-time analysis of robust ERM over linear models (the direct analogue of Theorem 3.3). A similar learning rate schedule appears in the works of Gunasekar et al. [2018a], Lyu and Li [2020]. To allow a fair comparison between the two algorithms, we stop their execution when they reach the same training loss value[3] ($10^{-3}$). We start from the all-zero, $\mathbf{w} = 0$, initialization.

**Diagonal neural networks** For the experiments with the diagonal linear network $f(\mathbf{x}; \mathbf{w}) = \langle \mathbf{w}, \mathbf{x} \rangle$, with $\mathbf{w} = \mathbf{u}_+^2 - \mathbf{u}_-^2$, we use a constant, small, learning rate $2 \times 10^{-3}$ and we adopt the same stopping criterion as with the "vanilla" linear models. We initialize $\mathbf{u}_+, \mathbf{u}_-$ with a constant value of $\frac{\alpha}{\sqrt{2d}}$ (where $\alpha$ is the initialization scale and $d$ the input dimension) which has been standard in prior works with diagonal networks. We set $\alpha = 10^{-3}$ to promote "feature" learning, i.e. to induce the implicit bias "faster" - see [Woodworth et al., 2020].

For all the runs with the various models/algorithms, we sample $d^2$ independent points and use them as a test set (one draw per dimension, i.e. same test dataset across the different values of $m$ and $\epsilon$). The maximum value of perturbation, $\epsilon^\star$, is estimated by running ERM with coordinate descent for $10^5$ iterations. Notice that this results to a different $\epsilon^\star$ for each different draw of the dataset. The robust test accuracy is efficiently calculated, since the adversarial points can be calculated in closed form for linear models. Our experimental protocol tried to ensure that we reach 100% (robust) train accuracy in all runs. This is true in all cases but the distributions with sparse data, where we found that for

---

[3]If the algorithm has not reached this value after $2 \times 10^5$ iterations, we stop at that epoch.

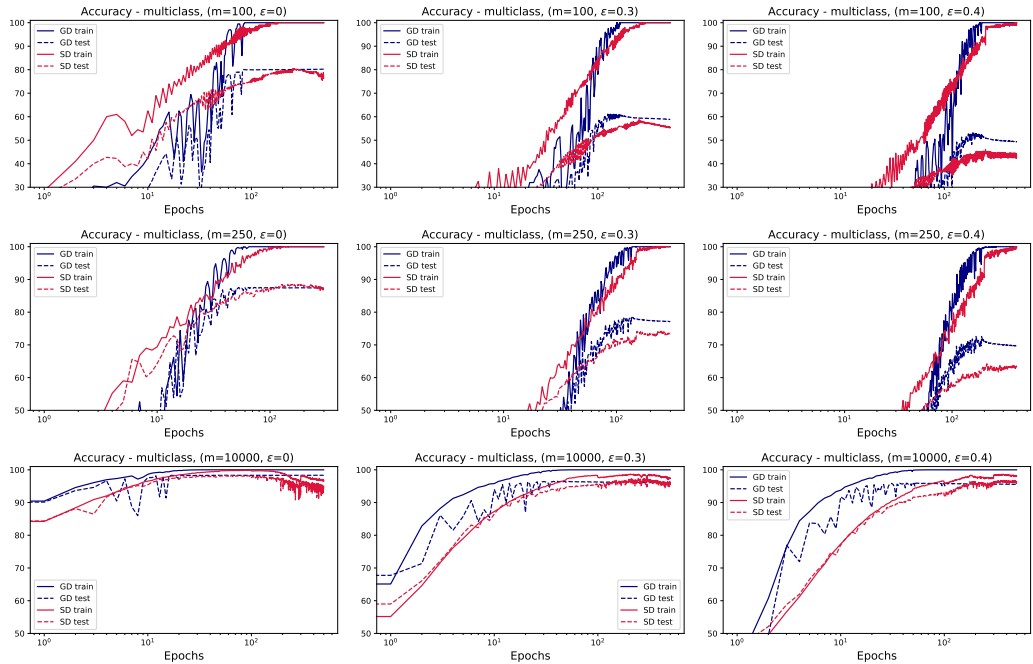

Figure 8: Training curves of CNNs trained on subsets of MNIST for various combinations of dataset size $m$ and perturbation magnitude $\epsilon$.

training datasets of cardinality $m = 2d$, the margin of the dataset was small and, thus, convergence was very slow. In these cases, the robust train accuracy approached 100%, but it was not exactly equal to it.

### F.2 Experiments with neural networks

In all the experiments with the various values of $\epsilon$ and $m$, we train with 3 random seeds which correspond to a different set of samples drawn from the full dataset and a different initialization.

**Fully Connected Neural Networks** We run GD and SD with the same set of hyperparameters; constant learning rate equal to $10^{-5}$, batch size equal to the whole available data (the amount varies across different experiments), and, when $\epsilon > 0$, we estimate robustness by calculating perturbations with (projected) gradient descent (PGD). We use 10 iterations of PGD with step size $\alpha = \frac{\epsilon}{5}$. The initialization of the networks is scaled down by a factor of $10^{-2}$ to promote feature learning and faster margin maximization. We use the exponential loss.

**Convolutional Neural Networks** We use a standard architecture from [Madry et al., 2018] consisting of convolutional, max-pooling and fully connected layers. The fully connected layers contain biases, so the network is not homogeneous. We start from PyTorch's default initialization. We used a constant learning rate equal to $0.1$ for GD and equal to $0.0001$ for SD (SD would diverge during training with larger learning rates that we tried). In Figure 3 (right) in the main text, we report the difference between accuracies obtained after convergence of train error to 0. In order to standardize the evaluation of the two algorithms, the reporting time corresponds to the first epoch hitting a certain train loss threshold, i.e. $10^{-3}$. For more challenging training regimes (i.e. large $\epsilon$), this threshold was set larger ($10^{-1}$), as we found it difficult to optimize to very small train loss values. In the experiments with CNNs, we used the cross entropy loss during training. When $\epsilon > 0$, we use 10 iterations of PGD with step size $\alpha = \frac{\epsilon}{5}$.

