# OpenReview forum: "The Price of Implicit Bias in Adversarially Robust Generalization"
_NeurIPS.cc/2024/Conference — NeurIPS 2024 poster_

### Official Review · Reviewer_ZNCk · 2024-07-08

**Soundness:** 3
**Presentation:** 3
**Contribution:** 2
**Rating:** 5
**Confidence:** 4

**Summary:**

This paper studies the generalization gap of robust empirical risk minimization for linear regression. The paper shows that the choice optimization algorithm or architecture affects the generalization gap of the trained linear model. In particular, a steepest descent algorithm w.r.t. $l_p$ norm finds the minimum $l_p$ norm interpolating solution on linearly separable data; a reparametrization of the linear model into a two-layer diagonal linear network has a bias toward minimum $l_1$ norm solution.

**Strengths:**

1. Connections between implicit bias (of optimization algorithm and of architecture) and adversarial robustness.
2. Interesting discussion on the optimal regularization for robust ERM w.r.t. $l_\infty$.
3. Experiments are through and well presented.

**Weaknesses:**

1. The message in this paper is delivered but the supporting argument is rather incomplete:
* Section 2.1 highlights the need for an optimal regularization given specific data assumption and threat model, but the discussion is primarily for $l_\infty$ threat model.
* Section 3.2 discusses how a diagonal linear network has a bias toward minimum $l_1$ solution, but the connection is not formal (the author acknowledges it in Remark 3.9).
2. The technical contribution is minor in my opinion. The ERM counterparts of the results in Section 3 are well-known and extensively studied, and extending them to robust ERM is more or less straightforward.

Minor comments:
1. Corollary 3.5 refers to equation (8), which has $p^*$ as the conjugate of $p$, yet the corollary itself contains another $p^*$, it is confusing whether they are the same $p^*$.
2. Referring steepest descent w.r.t. $l_1$ as "coordinate descent" is confusing. Generally, coordinate descent chooses the coordinate to be updated in a cyclic or random order. I understand there is a variation of CD that picks the coordinate with the largest gradient component, but plainly using CD may let the reader think of the more standard CD algorithm.

**Questions:**

See Weakness

**Limitations:**

See Weakness

---

> ### Author Rebuttal · Authors · 2024-07-31
>
> Thank you very much for taking the time to review our paper and for your positive evaluation of our work.
> First, let us correct a small mistake on your summary of our work (since reviews might become public in the future and readers might get confused):
> > This paper studies the generalization gap of robust empirical risk minimization for linear regression.
>
> We do not study regression problems, but classification with linear (as a function of the input) models. Experiments contain results with non-linear models.
>
>
> We address your questions:
>
>
> > Section 2.1 highlights the need for an optimal regularization given specific data assumption and threat model, but the discussion is primarily for $\ell_\infty$ threat model.
>
> Indeed, we chose to focus on the $\ell_\infty$ case in Section 2, since (a) this type has received much attention in practical applications and (b) it illustrates clearly the main message of our paper. Gradient descent is well positioned for $\ell_2$ perturbations in linear models, so there is not much value to study this case in detail from a generalization perspective. Please note that it has often been the case for theoretical studies on robustness in the past to only focus on one type of perturbation [1, 2]. In Section 3, however, our optimization results cover a general case of $\ell_p$ perturbations. Finally, in the experiments of Section 4.2 where we consider different classes of non-linear networks, we effectively study different kinds of threat models (since different feature extractors map $\ell_\infty$ perturbations to different spaces).
>
> > Section 3.2 discusses how a diagonal linear network has a bias toward minimum $\ell_1$ solution, but the connection is not formal (the author acknowledges it in Remark 3.9).
>
> We would like to clarify that Corollary 3.7 and Proposition 3.8 and their corrresponding proofs in the Appendix are formal and rigorous. However, as you point out, we can only show that we get convergence to a stationary point in Corollary 3.7. As it is often the case in results in this area (see for instance [3]), a full characterization of a limiting point as local optimum is elusive, so we often end up with characterization as first-order stationary points. It is unclear whether we can obtain stronger results here due to the non-smoothness of the adversarial objective, yet we are optimistic about it and we hope that future work can address this.
>
> > The technical contribution is minor in my opinion. The ERM counterparts of the results in Section 3 are well-known and extensively studied, and extending them to robust ERM is more or less straightforward.
>
> The optimization results are not trivial, since existing results for ERM, such as the ones found in [4], cannot be directly extended for the robust objective. In particular, Lemma 10 in pg. 18 in [4] cannot be generalized and this was a starting challenge that we faced. Furthermore, previous results on the implicit bias of gradient descent in robust ERM with linear models contain inconsistencies and unjustified steps [5]; the usage of Taylor's Theorem, as well as the bound on the sharpness of the Hessian in pg. 12 in that paper, appear to be wrong and it is technically non-trivial to sidestep these challenges which are due to the non-smoothness of the loss. As a result, we chose to analyze steepest flow for the robust objective, which has not been defined, let alone analyzed before. Particularly, Lemma C.2 uses a completely different idea than [4] and, since it is a general result about steepest descent/flow, it is interesting and beautiful in its own right.
>
> > Corollary 3.5 refers to equation (8), which has $p^\star$ as the conjugate of $p$, yet the corollary itself contains another $p^\star$, it is confusing whether they are the same $p^\star$.
>
> Thank you for this comment. They are the same $p^\star$.
>
> > Referring steepest descent w.r.t.  as "coordinate descent" is confusing. Generally, coordinate descent chooses the coordinate to be updated in a cyclic or random order. I understand there is a variation of CD that picks the coordinate with the largest gradient component, but plainly using CD may let the reader think of the more standard CD algorithm.
>
> Thank you for the suggestion. It is a matter of definitions and conventions. We followed the language of *Convex Optimization,
> Stephen Boyd and Lieven Vandenberghe* [6] (Section 9.4.2), which is one of the main references in Convex Optimization. We will add a note in the main text and in the appendix, stating that the algorithm should not be confused with other variations of CD. Thank you!
>
> Please let us know if our response addressed your concerns and we hope that you would consider raising your score if this is the case. Thank you!
>
>
> [1]. The Double-Edged Sword of Implicit Bias: Generalization vs. Robustness in ReLU Networks. Spencer Frei, Gal Vardi, Peter L. Bartlett, Nathan Srebro.
>
> [2]. Rademacher Complexity for Adversarially Robust Generalization. Dong Yin, Kannan Ramchandran, Peter Bartlett.
>
> [3]. Gradient Descent Maximizes the Margin of Homogeneous Neural Networks. Kaifeng Lyu, Jian Li.
>
> [4]. Characterizing Implicit Bias in Terms of Optimization Geometry. Suriya Gunasekar, Jason Lee, Daniel Soudry, Nathan Srebro.
>
> [5]. Yan Li, Ethan X. Fang, Huan Xu, and Tuo Zhao. Implicit bias of gradient descent based adversarial training on separable data.
>
> [6]. Convex Optimization. Stephen Boyd and Lieven Vandenberghe.

---

### Official Review · Reviewer_Pp2c · 2024-07-09

**Soundness:** 4
**Presentation:** 3
**Contribution:** 2
**Rating:** 5
**Confidence:** 3

**Summary:**

The paper studies the implicit bias of robust Empirical Risk Minimization (ERM) and its connection with robust generalization. In regularized classification, the authors discuss the choice of regularization for a given perturbation set to improve robust generalization. In the unregularized setting, they study the implicit bias of steepest descent when applying it to the worst-case exponential loss in scenarios where the data is separable. They investigate two architectures: linear models and diagonal neural networks.

**Strengths:**

1. The paper is well-written, and its contributions are well-explained.
 2. The difference between the convergence of Gradient Descent in linear models and diagonal neural networks with $\ell_{\infty}$ perturbations is a very interesting result.

**Weaknesses:**

In my opinion, a weakness of the paper is that while the authors engage in an interesting discussion in the technical sections, the results presented in the paper are not very insightful on their own:

1. The result presented in Section 2 is directly derived from Theorem 2.1, which is borrowed from prior works, and Rademacher Complexity.
 2. As the authors mention, the result of implicit bias in linear models is not surprising, and its proof is based on techniques from prior works.
3. The result of implicit bias in diagonal neural networks can be seen as a paraphrased theorem from prior work.

**Questions:**

Could the authors elaborate on the technical challenges they faced in proving their results, especially the result of implicit bias in linear models?

**Limitations:**

Yes

---

> ### Author Rebuttal · Authors · 2024-07-31
>
> Thank you very much for taking the time to review our work and help us improve it. We first reply to some of your points regarding the weaknesses of this paper:
>
> > As the authors mention, the result of implicit bias in linear models is not surprising, and its proof is based on techniques from prior works.
>
> While it is true that we can anticipate such a result for the implicit bias of steepest descent in robust ERM, we argue that the main novelty lies in asking the question in the first place ("*What is the effect of implicit bias of robust ERM in robust generalization?*") and in its connection with generalization bounds. There are no prior works that study these questions. Furthermore, please note that the proof is not a simple extension of previous results and it is not based directly on the techniques of [1] - see also our response to your question below.
>
> > The result of implicit bias in diagonal neural networks can be seen as a paraphrased theorem from prior work.
>
> We respectfully disagree. We leverage a result from prior work about robust ERM (Theorem 3.6) which characterizes the implicit bias of gradient descent for homogeneous networks in *parameter space*, while Corollary 3.7 and Proposition 3.8 establish the implicit bias of robust ERM in *function/predictor space*. This result is the first result of this kind for robust ERM. We are also confused about your previous point, since in Strength no.2 you complimented this result.
>
> Responding to your question,
>
> > Could the authors elaborate on the technical challenges they faced in proving their results, especially the result of implicit bias in linear models?
>
> The optimization results are not trivial, since existing results for ERM, such as the ones found in [1], cannot be directly extended for the robust objective. In particular, Lemma 10 in pg. 18 in [1] cannot be generalized and this was a starting challenge that we faced. Furthermore, previous results on the implicit bias of gradient descent in robust ERM with linear models contain inconsistencies and unjustified steps [2]; the usage of Taylor's Theorem, as well as the bound on the sharpness of the Hessian in pg. 12 in that paper, appear to be wrong and it is technically non-trivial to sidestep these challenges which are due to the non-smoothness of the loss. As a result, we chose to analyze steepest flow for the robust objective, which has not been defined, let alone analyzed before. Particularly, Lemma C.2 uses a completely different idea than [1] and, since it is a general result about steepest descent/flow, it is interesting and beautiful in its own right.
>
> More generally, we would like to defend the contributions of our paper (objecting to your score: "poor") by summarizing the most important ones:
> 1. We connect the implicit bias of optimization in robust ERM with the robust generalization error, and we show how and why implicit bias is more significant in robust ERM than in stardard ERM (this is where the term "price" comes from).
> 2. We paint a conceptual picture for the challenges faced in robust machine learning, which is rooted in foundational ideas of learning theory.
> 3. We rectify a misconception from prior work [3, 4] that asserted that regularization via the dual norm is always beneficial for robust generalization.
> 4. We provide rigorous results for linear models and diagonal neural networks against general $\ell_p$ adversarial perturbations.
> 5. We provide extensive experiments with synthetic data that validate the theoretical claims.
> 6. We provide several experiments with neural networks in image classification settings with gradient descent and sign gradient descent, and experimentally identify and establish a new connection between robust generalization gap and optimization geometry.
>
> We believe that the price of implicit bias in robust generalization is a new interesting phenomenon in robust machine learning and we believe there are many avenues for future work which can be inspired by this work.
>
> We hope that succintly summarizing the contributions of our paper, while also elaborating on some technical challenges that we faced by anwering your question could make you reconsider your evaluation of our work. We would be happy to include some discussion on the technical challenges, if you think this would benefit the paper. Thank you very much!
>
>
> [1]. Characterizing Implicit Bias in Terms of Optimization Geometry. Suriya Gunasekar, Jason Lee, Daniel Soudry, Nathan Srebro.
>
> [2]. Yan Li, Ethan X. Fang, Huan Xu, and Tuo Zhao. Implicit bias of gradient descent based adversarial training on separable data.
>
> [3]. Rademacher Complexity for Adversarially Robust Generalization. Dong Yin, Kannan Ramchandran, Peter Bartlett.
>
> [4]. Adversarial Learning Guarantees for Linear Hypotheses and Neural Networks. Pranjal Awasthi, Natalie Frank, Mehryar Mohri.

---

> ### Comment · Reviewer_Pp2c · 2024-08-11
>
> Thank you for the answers and clarifications.
>
> I still believe that the technical contributions are limited and partially involve extending results from previous works to the robust objective.
>
> However, I agree that one of the major contributions of the paper is highlighting the phenomenon of the price of implicit bias in robust machine learning.
>
> As a result, I have revised my score accordingly.

---

> > ### Author Response · Authors · 2024-08-14
> >
> > Thank you!

---

### Official Review · Reviewer_mpQX · 2024-07-10

**Soundness:** 4
**Presentation:** 3
**Contribution:** 3
**Rating:** 7
**Confidence:** 2

**Summary:**

In this paper, the authors study the issue of large generalization gap with Robust ERM objective, they connect this with the implicit bias of optimization (including architecture and the optimization algorithm). The findings suggest that optimizing models for robust generalization is challenging since it is hard to do the right capacity control for robust machine learning.

**Strengths:**

- The paper has in-depth investigations into how does the choice of regularization norm affect the generalization ability of the model w.r.t. sparsity of data, optimization algorithm and choice of architecture
- The authors validate their findings in both linear models and NN

**Weaknesses:**

- The theory studies might be still too limited

**Questions:**

See weakness

**Limitations:**

See weakness

---

> ### Author Rebuttal · Authors · 2024-07-31
>
> Thank you for reviewing & positively assessing our work and for highlighting its strengths. We address your only concern:
>
> > The theory studies might be still too limited
>
> This paper is the first to consider the connection between the implicit bias of optimization in adversarial training/robust ERM and robust generalization and it is also the first that identifies implicit bias as a main source of many challenges in robust machine learning (such as the large generalization gap, robustness/accuracy tradeoff, etc.). As a result, we believe it is natural that we began our theoretical analysis with the simplest models that permit this. Please note that stronger theoretical results would have required tight generalization bounds for more complicated classes of models (such as neural networks) and this is a central question in deep learning theory, even in the absence of adversarial perturbations - see, for instance, [1]. It could be interesting to derive a more general result for the case of one hidden layer ReLU neural networks, where a tight (robust) generalization bound exists (Theorem 10 in [2]). However, obtaining a characterization of the implicit bias of robust ERM with various optimization algorithms (in *predictor space*) is highly non-trivial for this class of models and would require many new techniques and generalization of previous results on ERM from e.g. the work of Savarese et al. [3]. We are actively working in this direction for a follow-up study.
> **Furthermore**, our experimental results on neural networks suggest that the phenomena that we identify are more general and we are happy that you recognized this in your evaluation of the strengths of this work. We would argue that the theoretical part of our study is not limited (since it is rich in new ideas) and we would be happy to listen to any suggestions that you might have on directions that might be worth developing further.
>
> [1]. Spectrally-normalized margin bounds for neural networks. Peter Bartlett, Dylan J. Foster, Matus Telgarsky.
>
> [2]. Adversarial Learning Guarantees for Linear Hypotheses and Neural Networks. Pranjal Awasthi, Natalie Frank, Mehryar Mohri.
>
> [3]. How do infinite width bounded norm networks look in function space? Pedro Savarese, Itay Evron, Daniel Soudry, Nathan Srebro.

---

> > ### Comment · Reviewer_mpQX · 2024-08-09
> >
> > Thank you for your detailed response and I will keep my positive score

---

> > > ### Author Response · Authors · 2024-08-14
> > >
> > > Thank you!

---

### Official Review · Reviewer_ey2u · 2024-07-12

**Soundness:** 3
**Presentation:** 3
**Contribution:** 3
**Rating:** 7
**Confidence:** 3

**Summary:**

This paper explores a linear classification scenario, investigating the factors that contribute to the gap between empirical adversarial risk and expected adversarial risk. Furthermore, they discuss which type of regularization should be applied in different cases. There are also simulations results to support their points.

**Strengths:**

1.The paper is well-written with a clear structure. Motivations are well-explained on why the authors study the problem and the contributions of this study are well discussed. The analysis for the theoretical results are helpful in understanding. Overall, it is easy to follow the logic and flow of the paper.

2.Theoretical results are solid and well-organized. The authors made the theoretical settings clear.

3.Empirical results support the theoretical findings.

**Weaknesses:**

1.In Theorem 2.1, it is not clear whether the constant $\rho$ has influences on other constants shown in the theorem.

2.While these results mainly focus on the gap between empirical adversarial risk and expected adversarial risk, maybe the discussions about their influences on expected risk and empirical adversarial risk are lacked.

3.As Theorem 3.3 focuses on steepest gradient dynamics on linear model, and Theorem 3.6 is about gradient flow on diagonal model, from my side, it is better to add a result about steepest gradient dynamics on diagonal model to make the analysis more sufficient.

**Questions:**

See weakness.

**Limitations:**

No negative social impact.

---

> ### Author Rebuttal · Authors · 2024-07-31
>
> Thank you very much for your critical and positive evaluation of our work. We respond to your questions:
>
> 1. > In Theorem 2.1, it is not clear whether the constant $\rho$ has influences on other constants shown in the theorem.
>
>     Thank you for the comment. The empirical margin $\hat{\rho}$ appears in the statement of Proposition 2.2 due to a typo, and it does not affect the bound in eq. 7, nor does it influence any constants as presented there. In fact, it is not used in the proof in the Appendix. We can obtain more refined, data-dependent versions of this bound by leveraging the empirical margin $\hat{\rho}$, but they do not offer any insights related to our discussion in Section 2, and we opted not to include such a result for the sake of brevity, as it could have misdirected the focus. See also our remark in lines 692-693 in the proof in the Appendix, where we address this issue and point the interested reader to a standard reference that discusses the role of $\hat{\rho}$ in generalization bounds. We removed the constant from the statement in lines 127-128, as well as from line 147, where it also appeared due to a typo. Thank you!
>
>
> 2. > While these results mainly focus on the gap between empirical adversarial risk and expected adversarial risk, maybe the discussions about their influences on expected risk and empirical adversarial risk are lacked.
>
>     Thank you for this very interesting suggestion (we assume that you meant to say "expected risk and empirical risk", instead of "expected risk and empirical adversarial risk")! In short, our search for robust predictors can actually harm the standard generalization error (error measured without perturbations), **if** the type of the perturbation is not "aligned" with the type of data. For example, in the case of *Dense, Dense* data in pg. 4, we expect from the generalization bounds to see a tradeoff between robustness and accuracy. The reason is that small $\ell_2$ norm solutions are preferred for standard generalization, while small $\ell_1$ solutions are better for robustness. This can also be observed in the experiments. We measure the standard generalization error ($\epsilon=0$, regardless of the value of $\epsilon$ used during training) and we produce a figure similar to Figure 2, consisting of heatmaps with the average difference between the performance of gradient descent (GD) and coordinate descent (CD). The figure can be accessed through the following link: https://ibb.co/sw7TWMD. Indeed, we observe that, for example, when $k_{\mathcal{X}}=k_{\mathcal{W}}=d$ (*Dense, Dense*) and we train for large perturbations, $\epsilon = \frac{\epsilon^\star}{2}$ (bottom right corner in top left subplot), GD generalizes better than CD in terms of standard generalization error. If we contrast this with the same datapoint in Figure 2 (where we measure the robust error), we see that there is a tradeoff between robustness and accuracy, since CD has better robust generalization. However, this is not always the case; for example for *Sparse, Dense* data, no such tradeoff is observed. This provides a more nuanced understanding of this well documented tradeoff [1] and suggests that the implicit bias of optimization in robust ERM is at the heart of this tradeoff as well. The focus of the paper was on robust generalization, but we agree with you that a discussion of standard generalization could be interesting in this context, space permitting. We will include the above results in the experiments section and add some discussion in Section 2 after Proposition 2.2. Thank you!
>
> 3. > As Theorem 3.3 focuses on steepest gradient dynamics on linear model, and Theorem 3.6 is about gradient flow on diagonal model, from my side, it is better to add a result about steepest gradient dynamics on diagonal model to make the analysis more sufficient.
>
>     The reason we opted to include results for gradient flow is essentially the non-smoothness of the robust loss for general $\ell_p$ perturbations, which complicates the technical arguments. In fact, as far as we understand, many proofs that have appeared in prior works concerning the implicit bias of gradient descent in robust ERM contain unjustified steps and non-rigorous parts and it is unclear how to rectify them, without many additional simplifying assumptions. Take, for instance, the proof of Theorem 3.1 in [2], which is about the implicit bias of gradient descent applied on the robust loss with linear models. The usage of Taylor's Theorem, as well as the bound on the sharpness of the Hessian on pg. 12 appear to be wrong and it is technically non-trivial to sidestep these challenges which are due to the non-smoothness of the loss. While providing rates of convergence is very interesting in its own right, such results do not affect our conclusions on the importance of implicit bias in the robust generalization of models.
>
>
> We would like to thank you once again for reviewing our work and helping us improve its presentation, especially with regard to weakness no.2. Please let us know if you have any more questions. If there are no outstanding concerns, we would kindly ask you to consider raising your score which would substantially help in reaching a reviewer consensus. Thank you.
>
> [1]. Tsipras, D., Santurkar, S., Engstrom, L., Turner, A., and Madry, A. Robustness may be at odds with accuracy.
>
> [2]. Yan Li, Ethan X. Fang, Huan Xu, and Tuo Zhao. Implicit bias of gradient descent based adversarial training on separable data.

---

> > ### Comment · Reviewer_ey2u · 2024-08-12
> > **Response to rebuttal**
> >
> > Many thanks for addressing my questions. I'll raise my score to 7.

---

> > > ### Author Response · Authors · 2024-08-14
> > >
> > > Thank you!

---

### Decision · Program_Chairs · 2024-09-25

**Decision:**

Accept (poster)

**Comment:**

The paper studies how the implicit bias of optimization in robust empirical risk minimization (robust ERM) influences robust generalization under adversarial perturbations. It identifies two key factors affecting robustness: the optimization algorithm and the model architecture. The study is supported by simulations with synthetic data and experiments with deep neural networks. The reviewers were satisfied with the responses provided during the rebuttal and found the paper interesting. The authors are encouraged to incorporate the comments and feedback received to strengthen their work.